# Fully Polynomial-Time Randomized Approximation Schemes for Global Optimization of High-Dimensional Minimax Concave Penalized Generalized Linear Models

## Abstract

Global solutions to high-dimensional sparse estimation problems with a folded concave penalty (FCP) have been shown to be statistically desirable but are strongly NP-hard to compute which implies the non-existence of pseudo-polynomial time global optimization schemes in the worst case. This paper shows that, with high probability, a global solution to generalized linear models with minimax concave penalty (MCP), a specific type of FCP, coincides with a stationary point characterized by the significant subspace second order necessary conditions ($S^3$ONC). Given that the desired $S^3$ONC solution admits a fully polynomial-time randomized approximation scheme (FPRAS), we are able to demonstrate the existence of an FPRAS for this strongly NP-hard problem. We further demonstrate two versions of the FPRAS for generating the desired $S^3$ONC solutions. One follows the paradigm of an interior point trust region algorithm and the other is the well-studied local linear approximation (LLA). Our analysis thus provides new techniques for global optimization of certain NP-Hard problems and new insights on the effectiveness of LLA.

## 1 Introduction

This paper concerns global optimization of a folded concave penalized formulation of high-dimensional learning generalized linear models, which belongs to statistical/machine learning problems such that the number of dimensions (or number of fitting parameters) $p$ is (much) larger than the number of samples $n$. This type of problem has recently become very common in a variety of engineering and scientific applications (Fan et al., 2014a; Fan & Li, 2006) including computational biology, speech recognition and image processing (Huang et al., 2016; Algamal & Lee, 2015; Xing et al., 2001; Palmer & Ostendorf, 2001; Bagaria et al., 2012; Paul et al., 2013). Globally minimal solutions to such a nonconvex learning formulation have been shown effective to guarantee desirable statistical performance in order to address high dimensionality (Zhang & Zhang, 2012). Nonetheless, generating a global solution admits no pseudo polynomial-time algorithm, unless "P = NP"; Indeed, global optimality is shown strongly NP-hard to achieve by (Chen et al., 2017; Huo & Chen, 2010) while Chen et al. (2014) show similar results for several related problems in regularized minimization. In contrast to the existing pessimistic result, we derive herein a fully polynomial-time randomized approximation scheme (FPRAS, as defined in 4) that theoretically ensures global minimality to 2 with high probability.

Specifically, consider a high-dimensional generalized linear model (GLM) as follows. Let $X = (x_1, ..., x_n)^\intercal$ be the $n \times p$ design matrix with $x_i = (x_{i1}, ..., x_{ip})^\intercal$, $i = 1, ..., n$, and $Y = (y_1, ..., y_n)^\intercal$ be the $n$-dimensional response vector. We will assume the design matrix $X$ is fixed, while the mean of the response is given by $E[y_i] = \psi'(x_i^\top \beta^{true})$ for some known link function $\psi : \Theta \to \Re$, where $\Theta \subseteq \Re$ and $\beta^{true} = (\beta_1^{true}, ..., \beta_p^{true})$ is the unknown vector of true parameters of the model. Such a setup can be seen as a generalization of linear regression models with the link function allowing for nonlinear transformations that enable a more flexible approach to model estimation. The high-dimensional regression problem is to estimate $\beta^{true}$ given knowledge of $X$, $Y$, and $\psi$ in the undesirable scenario where $p \gg n > 0$. To that end, traditional statistical learning schemes

often resort to the following formulation:

$$\mathcal{L}(\beta) = \sum_{i=1}^{n} \ell(y_i, x_i, \beta) = \frac{1}{n} \sum_{i=1}^{n} [\psi(x_i^\mathsf{T} \beta) - y_i x_i^\mathsf{T} \beta], \tag{1}$$

which, according to traditional statistical theories, would result in over fitting in general under the high-dimensional setting.

To resolve over fitting, modern statistical theories favor a modified formulation as below:

$$\min_{\beta} \left[ \mathcal{Q}(\beta) := \mathcal{L}(\beta) + \sum_{j=1}^{p} P_\lambda(|\beta_j|) \right], \tag{2}$$

where $P_\lambda(|\cdot|)$ is sparsity-inducing regularization term that penalizes any nonzero dimensions in the minimizer, and $\lambda > 0$ is a tuning parameter. Under the assumption that the true regression parameter $\beta^{true}$ is sparse, a global optimizer to equation 2 has been shown effective to address over fitting for many choices of specific regularization functions $P_\lambda$. Indeed, one of the most successful choices of $P_\lambda$ is the much studied Lasso-based regularized (Tibshirani, 1996), aka, the $\ell_1$(-norm) penalty, which has been demonstrated to entail desirable statistical properties (Bickel et al., 2009; Negahban et al., 2012). Another admirable property of the Lasso is that, especially when applied to least squares linear regression, it yields an extremely tractable problem via a variety of algorithms (Friedman et al., 2008; 2010). However, per (Zhao & Yu, 2006; Fan & Li, 2001), Lasso is not selection consistent without a strong irrepresentable condition and may sometimes introduce non-trivial estimation bias.

As a popular alternative to Lasso, the folded concave penalty (FCP) was first introduced by (Fan & Li, 2001). There are mainstream examples of FCP functions, including the SCAD by (Fan & Li, 2001) and MCP by (Zhang, 2010). This paper will focus on MCP, defined as $P_\lambda(|t|) = \int_0^{|t|} \frac{(a\lambda - s)_+}{a} \, ds$ for some fixed parameter $a > 0$. In contrast to the Lasso, the FCP regulaizations achieve variable selection consistency non-contingent on the irrepresentable condition and is demonstrated to be unbiased (Fan & Li, 2001). Furthermore, (Zhang & Zhang, 2012) demonstrated that the global solution to the FCP-regularized formulation leads to desirable recovery of the oracle solution.

Nonetheless, FCP problems are significantly harder to solve than Lasso, the new penalty term moves the problem outside the realm of convex optimization, (Chen et al., 2017) even showed that any estimation problem with convex loss and folded concave regularization to be strongly NP-hard, ruling out the possibility of a pseudo-polynomial-time global optimization algorithm. (Liu et al., 2016) were seemingly the first to propose a global approach to the problem called MIPGO which reformulates the problem into a mixed integer program. Yet, the worst-case complexity of MIPGO is in exponential time.

Perhaps for this reason, current literature tends to focus on local algorithms for the FCP-regularized learning problems. The local quadratic approximation algorithm by (Fan & Li, 2001) is an example of a majorization minimization algorithm, an approach which is also related to the local linear approximation (LLA) algorithm proposed by (Zou & Li, 2008). LLA was further explored by (Fan et al., 2014b) showing the oracle property can be obtained with high probability despite the local approach. (Mazumder et al., 2011; Fan & Lv, 2011) demonstrate coordinate optimization approaches for FCP while (Wang et al., 2014) used an approximate regularization path-following algorithm to obtain the optimal convergence rate to statistically desirable local solution. (Wang et al., 2013) analyzed the CCCP algorithm and showed under certain conditions that it asymptotically finds the oracle estimator. (Liu et al., 2017) took an algorithm agnostic approach by analyzing local solutions satisfying second order KKT conditions and showed desirable statistical properties like recovering the oracle solution and sparisty. These results discussed above primarily relate to FCP-regularized linear regression, a special case of GLM where $\psi$ is specifically the identity function. For analyses which encompass GLM's with FCP regularizers, (Fan & Lv, 2011) showed that GLM's, even in ultra high dimensional variable selection problems, have oracle properties when using FCP regularization and demonstrated a coordinate optimization algorithm for finding local solutions. In the area of M-estimators, which is a further generalization of our estimation method beyond even GLMs, (Loh & Wainwright, 2013; Loh et al., 2017a) showed that under certain conditions all local solutions must be within statistical precision of the true parameter and its support while (Loh et al., 2017b) demonstrate a two-step algorithm involving composite gradient descent to find a local solution. Bian &

Chen (2014) demonstrate a optimality conditions for a class of nonconvex optimization problems using nonlipscitz regularization.

From the numerous results pertaining to local solution schemes above, our research question is why local solutions are repetitively successful. In other words, are there certain geometric properties of the learning formulation equation 2 with FCP that allow all local schemes to perform well independent of the specific designs of the algorithmic procedures? Our answer to this question is affirmative; we show herein that all local solutions within an efficiently achievable sub-level set are actually globally optimal. Those local solutions are characterized by the significant subspace second-order necessary conditions ($S^3$ONC) provable admit FPRAS's. The $S^3$ONC are weaker conditions than the standard second-order KKT conditions. As per this result, all $S^3$ONC-guaranteeing algorithms (which include a large spectrum of local algorithms) belong to the class of FPRAS's for global optimization of the strongly NP-hard FCP-based learning problem. We subsequently develop theories for two specific algorithms of this type: one gradient-based method and the other is the same as the LLA.

It is worth noting that (Zhang, 2010) provides conditions to establish the uniqueness of local solutions to FCP-based learning. When local solutions are unique, then any local optimization algorithms would ensure global optimality. However, a few critical assumptions are necessary to achieve the uniqueness result and, furthermore, many report numerical experiments, e.g., those in (Fan et al., 2014b; Liu et al., 2017; 2016; Fan & Li, 2001) indicate the non-uniqueness of local solutions, instead. In contrast, our results in this paper imposes only standard assumptions commonly shared by a flexible set of high-dimensional GLMs and are applicable even if the local solutions are non-unique. To our knowledge, this is the first geometric proof that global solutions coincide with pseudo-polynomial-time computable local solutions in an FCP-based regression formulation with high probability. The resulting algorithms are the first few FPRAS's to this problem.

Two works with notable relations to our own are Liu et al. (2017) and Liu & Ye (2019). The first applies a similar analytical framework to linear regression problems however our generalization to GLMs adds significant flexibility and it was unknown for their result that the oracle solution implies global optimality since it was only as of Zhang & Zhang (2012) that global optimality was known to potentially imply the oracle solution. Further, it is nontrivial to extend their existing result to global optimal results. (Liu & Ye, 2019) on the other hand is a more general setup than our own though the tradeoff is that our rate is better and $S^3$ONC solutions do not ensure global optimality to the in-sample training error for their setup.

The rest of this paper is organized as follows. Section 2 goes through specific problem assumptions and explains the $S^3$ONC. Section 3 contains our main result for global optimality and uses it to make additional claims for LLA. Section 4 contains numerical results to verify our theoretical findings. Section 5 provides concluding remarks.

In this paper we will use $\|\cdot\|_0$ to denote the number of nonzero entries, $|\cdot|$ to denote the $\ell_1$-norm if the argument is a vector, or cardinality if the argument is a set, $\|\cdot\|$ to denote the $\ell_2$-norm, $\|\cdot\|_{max}$ to denote the maximum norm and $\|\cdot\|_{min}$ to denote the absolute value of the entry with the smallest magnitude. $(\cdot)_+$ is used equivalently to $\max(0, \cdot)$. For any vector $v$, $v_{\mathcal{Q}}$ is intended as $(v_j : j \in \mathcal{Q})$. For any set $\mathcal{Q}$, we denote the complement as $\mathcal{Q}^c$. In particular, let $S$ be the true support set, that is, $\mathcal{S} := \{j : \beta_j^{true} \neq 0\}$ and its complement is $\mathcal{S}^c$. We occasionally use the term the "oracle solution" to refer to the solution $\beta^{oracle}$ defined as $\beta^{oracle} \in \underset{\beta\,:\,\beta_j=0,\,\forall j \notin \mathcal{S}}{\arg\min} \ \mathcal{L}(\beta)$. The oracle solution is a hypothetical solution which assumes prior knowledge of the true support set $S$ and thus can be considered a theoretical benchmark.

## 2 Setups, preliminaries, and assumptions

### 2.1 setups and assumptions

Our analysis will focus on sparse GLMs that have a fixed design matrix and satisfy the following assumptions:

(A1) Assume that

(i) $b_u \geq \psi''(x_i^\mathsf{T}\beta) \geq b_l > 0$ for all $x_i^\mathsf{T}\beta \in \Theta$;

(ii) There exists $K > 0$ such that the design matrix satisfies $\frac{1}{n}\|X_j\|^2 < K$ for all $j \in [p]$. Let the tuning parameter $a$ in $P_\lambda$ satisfy $K < (b_u a)^{-1}$.

(A2) The vector of residuals $W \in \Re^n$ such that $W := y - E[y|X]$ is subgaussian($\sigma$) which means it satisfies that

$P[|\langle W, v \rangle| \geq t] \leq 2\exp(-t^2/2\sigma^2)$, for all $v : \|v\| = 1$ and any $t > 0$;

(A3) There exists a sequence $\{r_d \geq 0 : d = 1, 2, ..., p\}$ such that the following are satisfied:

(i) For any $d_1, d_2 : 1 \leq d_1 \leq d_2 \leq p$, we have $r_{d_1} \geq r_{d_2}$;

(ii) There exists some $\tilde{p}^* : 2|\mathcal{S}| \leq \tilde{p}^* \leq p$ such that $r_{\tilde{p}^*} > 0$;

(iii) For all $d : 1 \leq d \leq p$, it holds that $n^{-1}\|X\beta\|^2 \geq r_d\|\beta\|^2$ for any $\beta \in \Re^p : \|\beta\|_0 \leq d$.

**Remark 1.** *Part (i) of (A1) states that our link function is both strongly convex and continuously differentiable; that is, the gradient being Lipschitz continuous. Many types of traditional GLM problems satisfy this constraint including those for normal (linear regression), categorical (logistic regression), binomial, gamma and Poisson distributions, although in some cases the mild assumption on the boundedness of $\Theta$ has to be made. Even though the original domain of the link function can be unbounded, one may still consider its bounded subset given that it contains the vector of true parameters. Part (ii) of (A1) can be assumed without loss of generality by normalizing the design matrix columns.*

**Remark 2.** *(A2) is a common assumption in the literature, for example (Negahban et al., 2012) (Wang et al., 2013).*

**Remark 3.** *Both (i) and (A2) are satisfied by a number of GLM setups, one example is linear regression. In such a setup, the response $Y$ takes a gaussian distribution, while the gradient of the link function (encoding $E[Y|X]$) is the identity. Note, it is difficult to have one without the other, since the loss formulation 1 is simply a log-likelihood maximization applied to a distribution within the class of exponential dispersion models (Jørgensen, 1987). Another classic example is logistic regression which is used for a Bernoulli or binomial distributed $Y$ along with a logit link function. It should be noted that we treat the matrix $X$ as fixed, so its generative distribution is not important to the analysis outside of whether it satisfies the assumptions and constraints mentioned. In the numerical experiments in Section 4, we use i.i.d. gaussian generation method since, as discussed for Definition 1, it means our design matrices will satisfy (A3) with high probability.*

**Remark 4.** *Assumption (A3) can be understood to be a lower bound on the eigenvalues for principal sub-matrices of $X^\mathsf{T}X$ of dimension $d \times d$ for all $d \in [p]$. For every $d : d \leq \tilde{p}^*$, the lower bounds are positive, meaning that the smallest eigenvalues of the $d \times d$ principal sub-matrices are assumed positive.*

*According to (Liu et al., 2017), Assumption (A3), for certain parameters, is provably a weaker condition than the restricted eigenvalue (RE) condition, as defined in Definition 1 below and first introduced by (Bickel et al., 2009) as a plausible assumption to allow for the desired recovery quality of Lasso. The RE condition is a common assumption in the high-dimensional learning literature, such as (Zhang et al., 2014) and (Fan et al., 2014b).*

**Definition 1.** *(RE condition (Zhang et al., 2014)) The matrix $X \in \Re^{n \times p}$ is said to satisfy the RE condition if, for some $r_e > 0$, it holds that $\frac{1}{n}\|X\delta\|^2 \geq r_e\|\delta\|^2$ for all $\delta \in \bigcup_{|\hat{S}|=s}\mathbb{C}(\hat{S})$ where $\mathbb{C}(\hat{S}) := \{\delta := (\delta_i) \in \Re^p : |\delta_{\hat{S}^c}| \leq 3|\delta_{\hat{S}}|\}, \delta_{\hat{S}^c} := (\delta_j : j \in \hat{S}^c), \text{ and } \delta_{\hat{S}} := (\delta_j : j \in \hat{S})$. Furthermore, the largest possible $r_e$ is said to be the restricted eigenvalue constant of $X$.*

*Random design matrices with with i.i.d. rows generated following subgaussian distributions as in (A2) have been shown to satisfy the RE condition with high probability (Zhou, 2009) while proposition 1 in Loh & Wainwright (2013) includes a proof that with high probability, restricted strong convexity (RSC) is satisfied for a setup equivalent to our own. Note that satisfaction RSC implies the RE condition above. Thus, within our setup, (A3) is also satisfied with high probability for our setting.*

## 2.2 Preliminaries on S³ONC

Our results focus on the S³ONC solutions, which has been formerly introduced by (Liu et al., 2017) in the special case of high-dimensional linear regression as a relaxation of the standard second-order

KKT conditions. The definition of S³ONC depends on the notion of first order necessary conditions (FONC) as below.

**Definition 2** (FONC). *A solution $\beta^*$ satisfies the first order necessary conditions (FONC) if*

$$0 \in 1/n \sum_{i=1}^{n} [\psi'(x_i^\intercal \beta^*) - y_i] x_i + \left( P_\lambda'(|\beta_j^*|)\partial(|\beta_j^*|), 1 \le j \le p \right) \tag{3}$$

*where $\partial(|\cdot|)$ denotes the subdifferential of $|\cdot|$.*

**Definition 3** (S³ONC). *A solution $\beta^*$ satisfies the significant subspace second-order necessary condition (S³ONC) if it satisfies FONC and for all $j \in \{j : \beta_j^* \neq 0\}$,*

$$\left. \frac{\partial^2 \mathcal{Q}(\beta)}{(\partial \beta_j)^2} \right|_{\beta=\beta^*} \ge 0 \tag{4}$$

*if the second derivative exists.*

**Remark 5.** *The S³ONC can be intuited as the second order necessary condition applied only to the dimensions where $\beta_j \neq 0$, i.e., the significant dimensions. Since the S³ONC is weaker than the standard second-order KKT conditions, any algorithm that guarantees the second-order KKT conditions can be used to obtain an S³ONC solution, by requiring a more stringent optimality condition, may be slower than necessary. One specifically S³ONC guaranteeing approach, presented in (Liu & Ye, 2019), utilizes an interior point trust region algorithm in order to guarantee an $S^3ONC$ solution in polynomial time. This is the scheme which will be used later in Section 4.*

## 3 MAIN RESULTS

We now present our theoretical results for global optimization of FCP penalized GLMs. All proofs can be found in the appendix. We will make use of a short-hand notation:

$$\beta^{Lasso} \in \arg\min \mathcal{L}(\beta) + \lambda|\beta|. \tag{5}$$

**Theorem 1.** *Suppose assumptions (A1), (A2), and (A3) with any $\tilde{p}^* : \tilde{p}^* \ge 2|\mathcal{S}|$. Let $\beta^*$ be an arbitrary $S^3ONC$ solution to equation 2 with $P_\lambda$ specified as the MCP. Assume that $\mathcal{Q}(\beta^*) \le \mathcal{Q}(\beta^{true}) + \Gamma$ for an arbitrary $\Gamma \ge 0$. (i) Let the sub-optimality gap satisfies $\Gamma < P_\lambda(a\lambda) - \frac{\sigma^2}{b_l n}\left(\tilde{p}^* + 2\sqrt{\tilde{p}^* t} + 2t\right)$; (ii) choose $P_\lambda(a\lambda) > \frac{\sigma^2}{2nb_l}(1 + 2\sqrt{t'} + 2t') + \frac{\frac{\sigma^2}{n}|\mathcal{S}|(1+2\sqrt{t'}+2t')+\Gamma b_l}{b_l(\tilde{p}^*-2|\mathcal{S}|+1)}$ and (iii) assume that the minimal signal strength satisfy*

$$\left\| \beta_{\mathcal{S}}^{true} \right\|_{\min} > \sqrt{\frac{8\sigma^2}{r_{\tilde{p}} b_l^2 n}\left(\tilde{p}^* + 2\sqrt{\tilde{p}^* t} + 2t\right) + \frac{8}{r_{\tilde{p}} b_l}\min\left\{\frac{\lambda^2}{r_{\tilde{p}}}|\mathcal{S}|, P_\lambda(a\lambda)|\mathcal{S}| + \Gamma\right\}}$$

*then the following two statements hold*

*(a)* $\beta^*$ *is an oracle solution with probability at least* $1 - \exp(-t + \tilde{p}^* \ln(\frac{pe}{\tilde{p}^*})) - \exp(-(\tilde{p}^* + 1)(t' - \ln p)) \cdot \frac{1-\exp(-(p-\tilde{p}^*)(t'-\ln p))}{1-\exp(-t'+\ln p)}.$

*(b)* $\beta^*$ *is both an oracle solution and an globally optimal solution to equation 2 with probability* $1 - 2\exp(-t + \tilde{p}^* \ln(\frac{pe}{\tilde{p}^*})) - 2\exp(-(\tilde{p}^* + 1)(t' - \ln p)) \cdot \frac{1-\exp(-(p-\tilde{p}^*)(t'-\ln p))}{1-\exp(-t'+\ln p)}.$

**Remark 6.** *Theorem 1 (especially in the second statement) is perhaps the first result that establishes a set of conditions for any $S^3ONC$ solution to be globally optimal with high probability. Further, this result is algorithm independent which allows for greater flexibility compared to most existing results as in (Loh et al., 2017b) and (Fan & Lv, 2011) which rely on a specific algorithm choice.*

**Remark 7.** *The second part follows quite easily from the first due to the uniqueness of $\beta^{oracle}$ as well as the fact that $\beta^{opt}$ must also be an $S^3ONC$ solution. Thus by applying the first part of the Theorem to $\beta^{opt}$ we are able to show that both our arbitrary $\beta^*$ and $\beta^{opt}$ coincide with the unique $\beta^{oracle}$.*

**Remark 8.** *The above constraints on $\Gamma$, $P_\lambda(a\lambda)$ and $\|\beta_\mathcal{S}^{true}\|_{min}$ may initially seem disparate but can all be converted to constraints on the sample size $n$ as is shown in Corollary 1 below. This is possible because $\Gamma$ can be bounded by some function of $n^{-\gamma}$ for some $\gamma > 0$. Given that, it can be seen that the lesser side of inequalities (i),(ii) and (iii) go to 0 as $n$ grows. Further discussion of how this is achieved for Corollary 1 can be found in Remark 12.*

**Corollary 1.** *Assume $\ln p \geq 1$, $b_l \leq 1$, and $s \geq 1$. Let $\beta^*$ be an $S^3ONC$ solution to equation 2. Let assumptions (A1), (A2), and the RE condition as defined in Definition 1 hold. Assume that $\mathcal{Q}(\beta^*) \leq \mathcal{Q}(\beta^{Lasso})$ almost surely, where $\beta^{Lasso}$ is the optimal solution to the Lasso problem with penalty coefficient $\lambda^{Lasso} = \sigma\sqrt{\frac{lnp}{n^{1-\gamma/2}}}$ where $\gamma \in [0,1]$ is an arbitrary scalar. Let $\lambda = \frac{\sigma}{r_e}\sqrt{\frac{\ln p}{n^{\gamma/2}}}$ and $a \in [.8, 1)$. There exist problem independent constants $C_1 > 0$, $C_2 > 0$ and $C_2 > 0$ such that if*

$$n > \max\left\{\frac{C_1}{b_l}, \left[C_2\frac{s}{b_l}\right]^{\frac{2}{1-\gamma}}, \left[C_3\frac{s\sigma^2\ln p}{\|\beta_\mathcal{S}^{true}\|_{min}^2 b_l^2 r_e^4}\right]^{2/\gamma}\right\} \qquad (6)$$

*Then $\beta^*$ is the global solution to 2 with probability at least $1 - C_4\exp(-C_5 sn^{\gamma/2}\ln p) - C_6\exp(-C_7 b_u n^{\gamma/2}\ln(p))$ for problem independent constants $C_4, C_5, C_6$ and $C_7$*

**Remark 9.** *Corollary 1 indicates that for $\gamma > 0$, the global optimal solution coincides with computable $S^3ONC$ solution with overwhelming probability given that the sample size meets certain requirements. It should specifically be noted that the relationship between $n$ and $p$ require only $\frac{\ln p}{n^{\gamma/2}} = O(1)$, which ensures the applicability to the high-dimensional setting even if $n \ll p$.*

**Remark 10.** *(Liu & Ye, 2019) has derived a gradient-based algorithm that provably ensures an $S^3ONC$ solution at pseudo-polynomial-time complexity. When $n$ is properly large, this pseudo-polynomial-time algorithm enables a straightforward design of an FPRAS for generating the global optimal solution as follows.*

---

**FPRAS:** A pseudo-polynomial-time algorithm that generates global optimum at high probability

---

**Step 1.** Initialize the parameters $\delta, \lambda, a, \hat{a}, k = 0$ and $\beta^{Lasso}$ by solving equation 5

**Step 2.** If Case 1: $|\beta_j^k| \in (0, a\lambda)$ for some $j = 1, ..., p$, then choose an arbitrary $\iota \in \{j : |\beta_j^k| \in (0, a\lambda)\}$ and solve

$$\beta_\iota^{k+1} \in \arg\min_\beta[\nabla\mathcal{L}(\beta^k)]_\iota \cdot \beta + P_\lambda(|\beta|)$$

$$s.t. \quad (\beta - \beta_\iota^k)^2 \leq \delta^2$$

and let $\beta_j^{k+1} = \beta_j^k \quad \forall j \neq \iota$. Go to Step 3.

Else Case 2: If —$\beta_j^k \notin (0, a\lambda)$ for all $j = 1, ..., p$ then for all $j = 1, ..., p$:
— If $\beta_j^k = 0$ then $\beta_j^{k+1} = \hat{a} \cdot \left[|[\nabla\mathcal{L}(\beta^k)]_j| - \lambda\right]_+ \cdot sign(-[\nabla\mathcal{L}(\beta^k)]_j)$.
— If $|\beta_j^k| \geq a\lambda$, then $\beta_j^{k+1} = \beta_j^k - \hat{a} \cdot [\nabla\mathcal{L}(\beta^k)]_j$. Go to Step 3.

**Step 3.** Algorithm stops if $|\beta_j^k| \notin (0, a\lambda)$ and $\|\beta_j^k - \beta_j^{k+1}\| < \delta$. Otherwise, let $k := k + 1$ and go to Step 2.

---

**Remark 11.** *Here, the above algorithm has iteration complexity of $\mathcal{O}\left((\mathcal{Q}(\beta^{Lasso}) - \mathcal{Q}(\beta^{opt})) \cdot \max\{(1/(2a) - b_u/2)^{-1}, 2b_u^{-1}, (1/a - b_u/2)^{-1}\} \cdot 1/\delta^2\right)$ for any $\gamma$-accuracy $S^3ONC$ solution. In this iteration complexity, all the quantities are verifiably upper bounded by a polynomial function of dimensionality $p$ and the desired accuracy $1/\gamma$. Furthermore, $\beta^{Lasso}$ is a solution to a convex problem, which be generated within polynomial time and the per–iteration problem admits a closed form, whose complexity is strongly in polynomial-time. Therefor, this algorithm is an FPRAS in generating an $S^3ONC$ (global) solution where the term FPRAS is defined in Definition 4.*

**Definition 4.** *(FPRAS) Let $f : L \to \Re_0^+$ be a function representing the optimal solution to a problem. Let $\mathcal{A}$ be a probabilistic algorithm, which takes as an input an instance of the problem $x \in L$, and a parameter $\epsilon > 0$. We call $\mathcal{A}$ a fully polynomial randomized approximation scheme, if it has the following properties*

- *the algorithm returns a value within the required precision with probability at least $1/2$.*

$$P\big[|\mathcal{A}(x,\epsilon) - f(x)| \leq \epsilon f(x)\big] > 1/2$$

- *the running time of $\mathcal{A}$ is polynomial in both the size of the input $x$ and the precision $1/\epsilon$*

*Note that the specific probability bound is fairly arbitrary as long as it is a constant probability bounded away from 1/2 (Vazirani, 2013).*

**Remark 12.** *We are able to remove $\Gamma$ from the result by bounding the performance difference between $\beta^{true}$ and $\beta^{lasso}$ using similar techniques as in Bickel et al. (2009). In order to use this bound for our $S^3ONC$ solution, we require that $\mathcal{Q}(\beta^*) \leq \mathcal{Q}(\beta^{Lasso})$. However, this can generally be obtained by initializing any $S^3ONC$ guaranteeing algorithm with $\beta^{Lasso}$ in a similar fashion to (Fan et al., 2014b) for LLA. The FPRAS above follows the same initialization scheme.*

**Remark 13.** *The above specification of values for $a, \lambda$ and $\lambda^{Lasso}$ can be thought of as examples rather than strict requirements. A closer examination of the proof for Corollary 1 will reveal that the values for $\lambda$ and $\lambda^{Lasso}$ can be chosen in a much more flexible fashion, though the corresponding values of $C_1$ through $C_7$ may be different for different combinations of $\lambda$ and $\lambda^{Lasso}$.*

The techniques used in the proof of Theorem 1 can be used to provide insights into other optimization schemes. As an example, we can apply the same analysis to the state-of-the-art FCP-based algorithm, LLA, using the framework in (Fan et al., 2014b) as a starting point.

---

**LLA:** local linear approximation.

---

**Step 1.** Set $k = 0$. Initialize the algorithm with $\beta^0 = \beta^{Lasso}$, where $\beta^{Lasso}$ is generated by solving equation 5. Let $N$ be the maximal iteration number.

**Step 2.** For all $k = 1, ..., N$, solve the following convex program to generate $\beta^{k+1}$:

$$\beta^{k+1} \in \arg\min \mathcal{L}(\beta) + \sum_{j \in [p]} P'_\lambda(|\beta^k_j|) \cdot |\beta_j|,$$

where $P'_\lambda$ is the first derivative of $P_\lambda$. Let $k := k + 1$.

---

We can show that in fact the LLA is another FPRAS that achieves the global optimal solution. The proof of this can be found in the appendix.

**Corollary 2.** *For problem equation 2. If $\|\beta^{true}_\mathcal{S}\|_{min} > (a + 1)\lambda$, $\lambda > \max\{\frac{3\lambda^{Lasso}s^{1/2}}{b_l r_e}, \frac{4\sigma\sqrt{s+2\sqrt{st_1}+2t_1}}{b_l(an/b_u)^{1/2}}, \frac{2\sigma\sqrt{s+2\sqrt{st_2}+2t_2}}{b_l\sqrt{nr_e}}\}$ and the RE condition in Definition 1 holds, the following holds.*

(a) *The LLA algorithm initialized with $\beta^{Lasso}$ converges to the oracle solution in two iterations with probability $1 - \phi_0 - \phi_1 - \phi_2$, where*

$$\phi_0 := P(\big\|\beta^{Lasso} - \beta^{true}\big\|_{max} > \lambda) \leq 2p\exp(\frac{-(\lambda^{Lasso})^2 nb_u a}{8\sigma^2}),$$

$$\phi_1 := P(\big\|\nabla_{S^c_{\tilde{p}}}\ell_n(\beta^{oracle})\big\|_{max} \geq \lambda) \leq (\frac{pe}{s})^s\exp(-t_1) + 2\exp(\frac{-\lambda^2 ab_u n}{8\sigma^2}),$$

$$\phi_2 := P(\big\|\beta^{oracle}_\mathcal{S}\big\|_{min} \leq a\lambda) \leq (\frac{pe}{s})^s\exp(-t_2),$$

(b) *If in addition (A1) and (A2) holds, while the parameters of $(a, \lambda)$ satisfy that $P_\lambda(a\lambda) > \frac{\sigma^2}{2nb_l}(1 + 2\sqrt{t_4} + 2t_4) + \frac{\frac{\sigma^2}{n}|\mathcal{S}|(1+2\sqrt{t_4}+2t_4)b_l}{b_l(\tilde{p}^* - 2|\mathcal{S}|+1)}$ and the and $P_\lambda(a\lambda) > \frac{\sigma^2}{b_l n}(\tilde{p}^* + 2\sqrt{\tilde{p}^*t_3} + 2t_3)$ and let the minimal signal strength satisfy $\|\beta^{true}_\mathcal{S}\|_{min} > \sqrt{\frac{8\sigma^2}{r_{\tilde{p}}b_l^2 n}(\tilde{p}^* + 2\sqrt{\tilde{p}^*t_3} + 2t_3) + \frac{8}{r_{\tilde{p}}b_l}\min\{\frac{\lambda^2}{r_{\tilde{p}}}|\mathcal{S}|, P_\lambda(a\lambda)|\mathcal{S}|\}}$ then the LLA algorithm initialized by $\beta^{Lasso}$ converges to the global solution in two iterations with probability at least $1 - \phi_0 - \phi_1 - \phi_2 - \phi_3$*

*where*

$$\phi_3 := P(B^{oracle} \neq B^{opt}) \leq \exp(-t_3 + \tilde{p}^* \ln(\frac{pe}{\tilde{p}^*}))$$

$$+ \exp(-(\tilde{p}^* + 1)(t_4 - \ln p)) \cdot \frac{1 - \exp(-(p - \tilde{p}^*)(t_4 - \ln p))}{1 - \exp(-t_4 + \ln p)}, \quad (7)$$

*and $t_1, t_2, t_3, t_4 > 0$ are arbitrary constants.*

**Remark 14.** *Since each iteration of the LLA solves a convex program, which can be done within polynomial-time. When $n$ is properly large, the above theorem then indicates that the LLA is another FPRAS in globally optimizing the FCP-based nonconvex formulation.*

## 4    NUMERICAL EXPERIMENTS

### 4.1    EXPERIMENTAL SETUP

We focus our tests on sparse logistic regression. Our problem and data are implemented in a similar way as (Fan et al., 2014b). We construct $\beta^{true}$ as below: Firstly, $\beta_{\mathcal{S}}^{true}$ is constructed randomly by choosing 10 elements of $\beta$ and choosing the magnitude of each to be a uniform value within $[1, 2]$. Each value is chosen to be negative with probability .5. Then, the remaining entries $\beta_{\mathcal{S}^c}^{true}$ are set to be 0. The design matrix $X \in \Re^{n \times p}$ is constructed by generating $n$ iterations of $x_i \sim N_p(0, \Sigma)$ where $\Sigma = (.5^{|j-j'|})_{p \times p}$. We then generate Y using a Bernoulli distribution where $P(y_i = 1) = (1 + e^{-x_i^{\mathsf{T}}\beta^{true}})^{-1}$. With this data, we train a logistic regression model by invoking Algorithm 1 in solving equation 2 with MCP for S$^3$ONC solutions initialized with Lasso implemented in Python 3. The tuning parameters $\lambda$ and $a$ are obtained by cross validation following (Fan et al., 2014b).

We would like to ascertain whether our FCP classifier, obtained using $S^3ONC$ methods, is actually the global optimal solution. We do this by taking each element of the FCP classifier and perturbing it to find a new potential solution. Each element's perturbation is independent and generated by a $N(0, 1/p^{1/2})$-random variable. We then check if this perturbed classifier has better FCP regularized performance on the training data than the FCP classifier. If not, we repeat until either a better solution is found, or until 2000 perturbations have been tried.

Additionally, we compare our solution's statistical performance to those of other popular regularization methods. Using the data generation method above, we obtain two sets of data, both with 100 samples. One set is for training the model, and the other is the test set for out-of-sample tests. We repeat the above process for 100 times to generate 100 training-and-test instances, each with 100 samples. We compare those trained using the method described above with Lasso solutions generated by the global minimizer to equation 5 and an estimator generated by solving equation 2 when $P_\lambda$ is substantiated by an $\ell_2$ penalty. The Lasso and $\ell_2$ classifiers are solved using the *scikit learn* python library.

We compare the above estimators in terms of statistical performance for both $\ell_1$ loss: $|\beta^* - \beta^{true}|$ and $\ell_2$ loss: $\|\beta^* - \beta^{true}\|$.

### 4.2    NUMERICAL RESULTS

Table 1: Percent of time FCP beat all perturbations

|  | $n = 100$ $p = 500$ | $n = 100$ $p = 1000$ | $n = 100$ $p = 1500$ | $n = 100$ $p = 2000$ |
|---|---|---|---|---|
| **% Best FCP** | 100% | 100% | 100% | 100% |

Table 1 contains the numbers from optimality analysis. This technique did not yield a single perturbed solution that could beat the FCP classifier obtained from the FPRAS in any of our thousands of iterations.

Table 2 shows the numerical results for the statistical performance measurements. We show the two performance measures for each of the three classifiers for tphree different problem types.

Table 2: Statistical performance of the four classifiers.

| Classifier | Measure | $n = 100,\ p = 1000$ | | $n = 100,\ p = 1500$ | | $n = 100,\ p = 2000$ | |
|------------|---------|------|----------|------|----------|------|----------|
| | | **Mean** | **Std. dev** | **Mean** | **Std. dev** | **Mean** | **Std. dev** |
| MCP | $\ell_1$ loss | 13.909907 | 1.471911 | 14.818059 | 1.698191 | 14.506226 | 1.480686 |
| | $\ell_2$ loss | 4.108019 | 0.320061 | 4.304993 | 0.374453 | 4.489184 | 0.399441 |
| Lasso | $\ell_1$ loss | 15.015975 | 1.039529 | 15.882654 | 1.29422 | 17.079414 | 1.545309 |
| | $\ell_2$ loss | 4.3255 | 0.25996 | 4.397969 | 0.326336 | 4.433467 | 0.362707 |
| $\ell_2$ penalty | $\ell_1$ loss | 22.211963 | 0.791955 | 26.026067 | 0.966091 | 28.485075 | 0.993699 |
| | $\ell_2$ loss | 4.734209 | 0.241683 | 4.738025 | 0.296726 | 4.755959 | 0.296746 |

As expected, the FCP classifier generally outperformed the lasso and $\ell_2$ classifiers. The margins are fairly thin between FCP and lasso, especially compared to the standard deviation. Other values of $n$ and $p$ were tried but the results generally followed the same pattern.

As a result we tentatively conclude that our numerical results align with our theoretical results though further testing of the global optimality probability would be valuable.

## 5 CONCLUSIONS

This paper investigates both the theoretical and empirical performance of FPRAS's on MCP regularized GLMs. Despite such a problem being strongly NP-Hard, we have shown two FPRAS that achieve global optimality. To our knowledge this is the first probability bound for global optimization of FCP regularized GLMs using an FPRAS. Further, the same technique can be used to extend other results in order to obtain global optimization bounds for a wide variety of problems.

Though this paper focuses on GLMs, further exploration will focus on the question whether similar results can be found for more general problem classes under weaker assumptions. High-dimensional M-estimation problems could potentially be a future avenue of investigation.

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

# A   APPENDIX

The Appendix is organized as below: Section A.1 presents the proofs for the main results, Sections A.2 and A.3 present central lemmata to be useful in Section A.1.

## A.1   PROOF OF MAIN RESULTS

A useful relationship in our proofs is that, for an $S^3ONC$ solution $\beta^*$ within $\{\beta^* : \mathcal{Q}(\beta^*) \leq \mathcal{Q}(\beta^{true}) + \Gamma\}$ for any $\Gamma \geq 0$, we have the following useful inequality under Assumption (A1):

$$\frac{b_l}{2n}\|X\delta^*\|^2 - \frac{1}{n}W^\intercal X\delta^* + \sum_{j\in\mathcal{S}}P_\lambda(|\beta_j^*|) \leq \sum_{j\in\mathcal{S}}P_\lambda(|\beta_j^{true}|) + \Gamma, \tag{8}$$

where $\delta^* = \beta^* - \beta^{true}$. This is obtained by invoking the strong convexity of $\psi$, which leads to $\psi(x_i^\intercal\beta^*) \geq \psi(x_i^\intercal\beta^{true}) + \psi'(x_i^\intercal\beta^{true})(x_i^\intercal\beta^* - x_i^\intercal\beta^{true}) + 0.5 \cdot b_l(x_i^\intercal\beta^* - x_i^\intercal\beta^{true})^2$.

*Proof of Theorem 1.* First, given our assumption that (A1) holds, that (i) $\tilde{p}^* \geq 2|\mathcal{S}|$, (ii) $\beta^*$ is $S^3ONC$ satisfying $\mathcal{Q}(\beta^*) \leq \mathcal{Q}(\beta^{true}) + \Gamma$ for some $\Gamma \geq 0$, and (iii) $P_\lambda(a\lambda) > \frac{\sigma^2}{2nb_l}(1 + 2\sqrt{t'} + 2t') + \frac{\frac{\sigma^2}{n}|\mathcal{S}|(1+2\sqrt{t'}+2t')+\Gamma b_l}{b_l(\tilde{p}^*+1-2|\mathcal{S}|)}$, we can apply Lemma 5 with $\tilde{p} = \tilde{p}^*$. This means that $\|\beta^* - \beta^{true}\|_0 \leq \tilde{p}^*$ with probability at least $1 - \exp(-(\tilde{p}^* + 1)(t' - \ln p)) \cdot \frac{1-\exp(-(p-\tilde{p}^*)(t'-\ln p))}{1-\exp(-t'+\ln p)}$. From this, given the additional assumption that (A3) holds, we can apply the second part of Lemma 4 with $\tilde{p} = \tilde{p}^*$ to get that for any $t > 0$, $\frac{1}{n}\|X(\beta^* - \beta^{true})\|^2 \leq \frac{8\sigma^2}{b_l^2 n}(\tilde{p}^* + 2\sqrt{\tilde{p}^* t} + 2t) + \frac{8}{b_l}\min\{\lambda^2(|\mathcal{S}| - \|\beta^*\|_0)r_{\tilde{p}^*}^{-1}, P_\lambda(a\lambda)(|\mathcal{S}| - \|\beta^*\|_0) + \Gamma\}$ holds with probability at least $1 - \exp(-t + \tilde{p}^*\ln(\frac{pe}{\tilde{p}^*}))$. Given that for 2 arbitrary sets $A$ and $B$

$$P(A \cap B) = P(B)P(A|B) = (1 - P(B^c))(1 - P(A^c|B))$$
$$= 1 - P(A^c|B) - P(B^c) + P(B^c)P(A^c|B)$$
$$= 1 - P(A^c|B) - P(B^c)(1 - P(A^c|B)) \geq 1 - P(A^c|B) - P(B^c)$$
$$(9)$$

Therefor they hold simultaneous with probability at least $1 - \exp(-t + \tilde{p}\ln(\frac{pe}{\tilde{p}^*})) - \exp(-(\tilde{p}^* + 1)(t' - \ln p)) \cdot \frac{1 - \exp(-(p - \tilde{p}^*)(t' - \ln p))}{1 - \exp(-t' + \ln p)}$.

The same sequence of arguments can be used to show that $\beta^{opt}$ also satisfies $\|\beta^{opt} - \beta^{true}\| \leq \tilde{p}^*$ and $\frac{1}{n}\|X(\beta^{opt} - \beta^{true})\|^2 \leq \frac{8\sigma^2}{b_l^2 n}\left(\tilde{p}^* + 2\sqrt{\tilde{p}^* t} + 2t\right) + \frac{8}{b_l}\min\{\lambda^2(|\mathcal{S}| - \|\beta^*\|_0)r_{\tilde{p}^*}^{-1}, P_\lambda(a\lambda)(|\mathcal{S}| - \|\beta^*\|_0) + \Gamma\}$ with the same probability. Using again the union bound and DeMorgan's law, we say $\beta^*$ and $\beta^{opt}$ satisfy the above conditions simultaneously with probability $1 - 2\exp(-t + \tilde{p}^*\ln(\frac{pe}{\tilde{p}^*})) - 2\exp(-(\tilde{p}^* + 1)(t' - \ln p)) \cdot \frac{1 - \exp(-(p - \tilde{p}^*)(t' - \ln p))}{1 - \exp(-t' + \ln p)}$. With this, our $\Gamma$ assumption and our minimal signal strength assumption, we can apply Lemma 6 to show that $\beta^* = \beta^{opt}$ with probability at least $1 - 2\exp(-t + \tilde{p}^*\ln(\frac{pe}{\tilde{p}^*})) - 2\exp(-(\tilde{p}^* + 1)(t' - \ln p)) \cdot \frac{1 - \exp(-(p - \tilde{p}^*)(t' - \ln p))}{1 - \exp(-t' + \ln p)}$. □

*Proof of Corollary 1.* First we need to bound $\Gamma$. In order to do this we use the lasso problem $\mathcal{Q}^{lasso}(\beta) = \sum_{i\in\mathcal{N}} \ell(\beta, x_i, y_i) + \sum_{j\in\mathcal{P}} \lambda^{lasso}|\beta_j|$ as well as the concavity of MCP over positive values to get the following 2 inequalities

$$\mathcal{Q}^{lasso}(\beta^{lasso}) \leq \mathcal{Q}^{lasso}(\beta^{true})$$
$$\sum_{i\in\mathcal{N}} \ell(\beta_j^{lasso}, x_i, y_i) - \ell(\beta_j^{true}, x_i, y_i) \leq \sum_{j\in\mathcal{P}} \lambda^{lasso}(|\beta_j^{true}| - |\beta_j^{lasso}|)$$
$$\leq \sum_{j\in\mathcal{P}} \lambda^{lasso}|\beta_j^{lasso} - \beta_j^{true}|$$
$$(10)$$

and

$$\sum_{j\in\mathcal{P}} P_\lambda(\beta_j^{true}) - \sum_{j\in\mathcal{P}} P_\lambda(\beta_j^{lasso}) \leq \sum_{j\in\mathcal{P}} P_\lambda'(\beta_j^{lasso})(|\beta_j^{true}| - |\beta_j^{lasso}|) \leq \sum_{j\in\mathcal{P}} \lambda|\beta_j^{lasso} - \beta_j^{true}|$$
$$(11)$$

We also need 2 results from the proof for $\phi_0$ in Corollary 2 which shows that both $|\delta_{\mathcal{S}_c}^\ell| \leq 3|\delta_{\mathcal{S}}^\ell|$ and $\frac{b_l}{n}\|X\delta^\ell\|^2 \leq 3\lambda^{lasso}|\delta_{\mathcal{S}}^\ell|$ conditional on $\mathcal{A}$ where $\delta^\ell = \beta^{lasso} - \beta^{true}$. Given our restricted eigenvalue assumption $\frac{\|X\delta^\ell\|^2}{n\|\delta^\ell\|^2} \geq r_e$, this can be used to show

$$|\delta^\ell| \leq 4|\delta_{\mathcal{S}}^\ell| \leq 4\sqrt{s}\frac{\|\delta_{\mathcal{S}}^\ell\|^2}{\|\delta_{\mathcal{S}}^\ell\|} \leq 4\sqrt{s}\frac{\|\delta^\ell\|^2}{\|\delta_{\mathcal{S}}^\ell\|}$$
$$\leq \frac{4\sqrt{s}}{r_e n}\frac{\|X\delta^\ell\|^2}{\|\delta_{\mathcal{S}}^\ell\|} \leq \frac{4\sqrt{s}}{r_e}\frac{3\lambda^{lasso}|\delta_{\mathcal{S}}^\ell|}{b_l\|\delta_{\mathcal{S}}^\ell\|} \leq \frac{4\sqrt{s}}{r_e}\frac{3\lambda^{lasso}\sqrt{s}\|\delta_{\mathcal{S}}^\ell\|}{b_l\|\delta_{\mathcal{S}}^\ell\|}$$
$$(12)$$

which means $|\delta^\ell| \leq \frac{12\lambda^{lasso}s}{b_l r_e}$ with conditional on $\mathcal{A}$ which occurs with probability at least $1 - 2p\exp(\frac{-(\lambda^{lasso})^2 n b_u a}{8\sigma^2})$

Finally we are able to bound gamma by combining the above

$$\Gamma \leq \mathcal{Q}(\beta^*) - \mathcal{Q}(\beta^{true}) \leq \mathcal{Q}(\beta^{lasso}) - \mathcal{Q}(\beta^{true}) \tag{13}$$

$$\leq \sum_{i \in \mathcal{N}} \ell(\beta_j^{lasso}, x_i, y_i) - \sum_{j \in \mathcal{P}} P_\lambda(\beta_j^{lasso}) - [\sum_{i \in \mathcal{N}} \ell(\beta_j^{true}, x_i, y_i) - \sum_{j \in \mathcal{P}} P_\lambda(\beta_j^{true})] \tag{14}$$

$$\leq \sum_{j \in \mathcal{P}} (\lambda^{lasso}|\beta_j^{lasso} - \beta_j^{true}| + \lambda|\beta_j^{lasso} - \beta_j^{true}|) \tag{15}$$

$$\leq (\lambda^{lasso} + \lambda)|\delta^\ell| \leq (\lambda^{lasso} + \lambda)\frac{12\lambda^{lasso}s}{b_l r_e} \tag{16}$$

Next consider the conditions necessary to apply Theorem 1. We have assumptions (A1) and (A2) and (A3) per our assumption that the RE condition holds combined with 7. That leaves the 3 requirements on $\Gamma$, $P_\lambda(a\lambda)$ and $\|\beta_{\mathcal{S}}^{true}\|_{min}$. We will convert each of these to inequalities on $n$

Utilizing 16 and substituting $\lambda = \frac{Q\sigma}{r_e}\sqrt{\frac{\ln p}{n^{\gamma/2}}}$ and $\lambda^{lasso} = \epsilon\sigma\sqrt{\frac{\ln p}{n^{1-\gamma/2}}}$, where $Q, \epsilon > 0$ are arbitrary constants, and setting $\tilde{p}^* = 4s$, $t = \tilde{p}^* n^{\gamma/2} \ln p$, $t' = n^{\gamma/2} \ln p$ we get the following

$$P_\lambda(a\lambda) > \frac{\sigma^2}{2nb_l}(1 + 2\sqrt{t'} + 2t') + \frac{\frac{\sigma^2}{2}s(1 + 2\sqrt{t'} + 2t') + \Gamma b_l}{b_l(\tilde{p}^* - 2s + 1)} \tag{17}$$

$$n > \frac{8 + 12\epsilon^2 + 12\epsilon Q}{b_l a Q^2} = C_1/b_l \tag{18}$$

$$\Gamma < P_\lambda(a\lambda) - \frac{\sigma^2}{b_l n}\left(\tilde{p}^* + 2\sqrt{\tilde{p}^* t} + 2t\right) \tag{19}$$

$$n > \left[\left(\frac{12\epsilon}{aQ} + \sqrt{\frac{20 + 12\epsilon^2}{aQ^2}}\right)\frac{s}{b_l}\right]^{\frac{2}{1-\gamma}} = \left[C_2\frac{s}{b_l}\right]^{\frac{2}{1-\gamma}} \tag{20}$$

$$\|\beta_{\mathcal{S}}^{true}\|_{min} > \sqrt{\frac{8\sigma^2}{r_{\tilde{p}}b_l^2 n}\left(\tilde{p}^* + 2\sqrt{\tilde{p}^* t} + 2t\right) + \frac{8}{r_{\tilde{p}}b_l}\min\{\frac{\lambda^2}{r_{\tilde{p}}}|\mathcal{S}|, P_\lambda(a\lambda)|\mathcal{S}| + \Gamma\}} \tag{21}$$

$$n > \left[(160 + 8Q^2)\frac{s\sigma^2 \ln p}{(\|\beta_{\mathcal{S}}^{true}\|_{min} r_{4s}b_l r_e)^2}\right]^{2/\gamma} = \left[C_3\frac{s\sigma^2 \ln p}{(\|\beta_{\mathcal{S}}^{true}\|_{min} r_{4s}b_l r_e)^2}\right]^{2/\gamma} \tag{22}$$

For some constants $C_1, C_2$ and $C_3$

We can then apply Theorem 1 (conditional on $\mathcal{A}$) substitute our values and simplify to get that $\beta^*$ is the global solution with probability at least

$$1 - 2\exp(-t + \tilde{p}^* \ln(\frac{pe}{\tilde{p}^*})) - 2\exp(-(\tilde{p}^* + 1)(t' - \ln p)) \cdot \left[\frac{1 - \exp(-(p - \tilde{p}^*)(t' - \ln p))}{1 - \exp(-t' + \ln p)}\right]$$

$$\geq 1 - 2\exp(-(n^{\gamma/2} - 1)4s \ln p) - 2\left[\sum_{k=1}^{p - \tilde{p}^*} \exp(-(\tilde{p}^* + k)(n^{\gamma/2} - 1) \ln p)\right]$$

$$\geq 1 - 2\exp(-(n^{\gamma/2} - 1)4s \ln p) - 2\exp(-[(4s + 1)(n^{\gamma/2} - 1) - 1] \ln p)$$

$$\geq 1 - C_4 \exp(-C_5 s n^{\gamma/2} \ln p) \tag{23}$$

We then use the same technique as in Theorem 1 to combine this number with the probability of $\mathcal{A}$ to get the final non-conditional probability that $\beta^*$ is the global solution with probability at least

$$\geq 1 - C_4 \exp(-C_5 sn^{\gamma/2} \ln p) - 2p \exp(\frac{-(\lambda^{lasso})^2 nb_u a}{8\sigma^2})$$

$$\geq 1 - C_4 \exp(-C_5 sn^{\gamma/2} \ln p) - 2 \exp(\frac{-(\epsilon^2 b_u a n^{\gamma/2} - 8) \ln p}{8}) \tag{24}$$

$$\geq 1 - C_4 \exp(-C_5 sn^{\gamma/2} \ln p) - C_6 \exp(-C_7 b_u n^{\gamma/2} \ln(p))$$

For some constants $C_4$, $C_5$ $C_6$ and $C_7$.

Note that these constants, as well as $C_1$, $C_2$ and $C_3$, are dependent only on the value of $a$ $Q$ and $\epsilon$, as far as problem dependencies are concerned. Thus given that $a$ $Q$ and $\epsilon$ are chosen to be any positive constant value, as in the statement of Corollary 1, $C_1$ through $C_7$ are problem independent, which is the desired result. $\qquad\square$

*Proof of Corollary 2.* The first result is simply Corollary 2 in (Fan et al., 2014b). If we initialize the LLA algorithm with $\beta^{lasso}$, the solution to LASSO using $\lambda^{lasso}$ as the LASSO constant, then the LLA algorithm converges to the oracle solution in 2 iterations with probability $1 - \phi_0 - \phi_1 - \phi_2$. However we still need to solve for the actual values of $\phi_0, \phi_1, \phi_2$ for GLM.

First consider $\phi_0 = P(\left\|\beta^{lasso} - \beta^{true}\right\|_{max} > a_0\lambda)$. Similar to Lemma B.1. in Bickel et al. (2009), to bound this we will start by noticing that for the lasso penalized loss function $\mathcal{Q}^{lasso}(\beta) = \sum_{i\in\mathcal{N}} l(\beta, x_i, y_i) + \lambda^{lasso}\sum_{j\in\mathcal{P}} |\beta_j|$ we have that $\mathcal{Q}^{lasso}(\beta^{lasso}) \leq \mathcal{Q}^{lasso}(\beta^{true})$. If we then let $\delta^\ell = \beta^{lasso} - \beta^{true}$ we can use the same tactic as in the derivation of 8 to get $\frac{b_l}{2n}\left\|X\delta^\ell\right\|^2 - \frac{1}{n}W^\intercal X\delta^\ell \leq \lambda^{lasso}\sum_{j\in\mathcal{P}} |\beta_j^{true}| - |\beta_j^{lasso}|$, which can then be rearranged to get.

$$\frac{b_l}{2n}\left\|X\delta^\ell\right\|^2 - \frac{1}{n}\sum_{j\in\mathcal{P}} |W^\intercal X_j||\delta_j'| \leq \lambda^{lasso}\sum_{j\in\mathcal{P}} |\beta_j^{true}| - |\beta_j^{lasso}|. \tag{25}$$

Next let $\mathcal{A} = \bigcap_{j\in\mathcal{P}}\{|\frac{1}{n}W^\intercal X_j| \leq \lambda^{lasso}/2\}$. We can combine this with 25 to get that $\frac{b_l}{2n}\left\|X\delta^\ell\right\|^2 + \lambda^{lasso}/2\sum_{j\in\mathcal{P}} |\beta_j^{lasso} - \beta_j^{true}| \leq \lambda^{lasso}\sum_{j\in\mathcal{P}} |\beta_j^{lasso} - \beta_j^{true}| + \lambda^{lasso}\sum_{j\in\mathcal{P}} |\beta_j^{true}| - |\beta_j^{lasso}|$ conditional on $\mathcal{A}$. From this notice that the right term goes to zero when $\beta_j^{true} = 0$ so we then have that $\frac{b_l}{2n}\left\|X\delta^\ell\right\|^2 + \lambda^{lasso}/2\sum_{j\in\mathcal{P}} |\beta_j^{lasso} - \beta_j^{true}| \leq \lambda^{lasso}\sum_{j\in\mathcal{S}} |\beta_j^{lasso} - \beta_j^{true}| + |\beta_j^{true}| - |\beta_j^{lasso}|$. Using the triangle inequality and the definition of $\delta^\ell$ we can simplify this to

$$\frac{b_l}{2n}\left\|X\delta^\ell\right\|^2 + \frac{\lambda^{lasso}}{2}|\delta^\ell| \leq 2\lambda^{lasso}|\delta_\mathcal{S}^\ell| \tag{26}$$

conditional on $\mathcal{A}$. By relaxing different parts of the equation, this can be further simplified to both $\frac{b_l}{n}\left\|X\delta^\ell\right\|^2 \leq 3\lambda^{lasso}|\delta_\mathcal{S}^\ell| \leq 3\lambda^{lasso}s^{1/2}||\delta_\mathcal{S}^\ell||_2$ and $|\delta_{\mathcal{S}_c}^\ell| \leq 3|\delta_\mathcal{S}^\ell|$. Note that the second of these shows that $\delta^\ell$ satisfies the constraint for the RE condition 1. Therefor we have that $\frac{\left\|X\delta^\ell\right\|^2}{n\|\delta^\ell\|^2} \geq r_e$. If this is combined with the first of the two equations, we can get that $\frac{1}{n^{1/2}}\left\|X\delta^\ell\right\| \leq \frac{3\lambda^{lasso}s^{1/2}}{b_l(r_e)^{1/2}}$ conditional on $\mathcal{A}$.

Next, using this we can show that conditional on $\mathcal{A}$ we have that $\left\|\delta^\ell\right\|_{max} \leq \left\|\delta^\ell\right\| \leq \left\|X\delta^\ell\right\|_2^2/(\left\|\delta^\ell\right\| nr_e) \leq \frac{3\lambda^{lasso}s^{1/2}}{b_l r_e} < a_0\lambda$ if $\lambda > \frac{3\lambda^{lasso}s^{1/2}}{b_l a_0 r_e}$. This is the inverse of the condition that defines $\phi_0$. Thus, we can bound $\phi_0$ with $\phi_0 \leq P(\mathcal{A}^c) = P(\bigcup_{j\in\mathcal{P}} |\frac{1}{n}W^\intercal X_j| > \lambda^{lasso}/2) = P(\bigcup_{j\in\mathcal{P}} |W^\intercal X_j|/\|X_j\| > n\lambda^{lasso}/(2\|X_j\|)) \leq pP(|\langle W, v\rangle| > \frac{\lambda^{lasso}n}{2\|X_j\|}) \leq pP(|\langle W, v\rangle| > \frac{\lambda^{lasso}(nb_u a)^{1/2}}{2}) \leq 2p\exp\frac{-(\lambda^{lasso})^2 nb_u a}{8\sigma^2}$ which uses both (A1)(ii) and (A2) as long as $\lambda > \frac{3\lambda^{lasso}s^{1/2}}{b_l a_0(r_e)}$ per (A2).

Next consider $\phi_1 = P(\left\|\nabla_{S_{\hat{p}}^c} \ell_n(\beta^{oracle})\right\|_{max} \geq a_1\lambda)$

$$\phi_1 = P(\left\| \nabla_{S_{\tilde{p}}^c} \ell_n(\beta^{oracle} \right\|_{max} \geq a_1\lambda) \tag{27}$$

$$= P(\exists j \in \mathcal{P} : |\nabla_j \ell_n(\beta^{oracle})| \geq a_1\lambda) \tag{28}$$

$$= P(\exists j \in \mathcal{P} : |\frac{1}{n}\sum_{i\in\mathcal{N}}[\psi'(x_i^\intercal\beta^{oracle})x_{i,j} - y_i x_{i,j}]| \geq a_1\lambda) \tag{29}$$

$$= P(\exists j \in \mathcal{P} : |\frac{1}{n}\sum_{i\in\mathcal{N}}[\psi'(x_i^\intercal\beta^{oracle})x_{i,j} - \psi'(x_i^\intercal\beta^{true})x_{i,j} + W_i x_{i,j}]| \geq a_1\lambda) \tag{30}$$

$$\leq P(\frac{1}{n}|X_j^\intercal(\psi'(X\beta^{oracle}) - \psi'(X\beta^{true}) + W)| \geq a_1\lambda) \tag{31}$$

$$\leq P(\frac{1}{n}|X_j^\intercal(\psi'(X\beta^{oracle}) - \psi'(X\beta^{true}))| + |W^\intercal X_j| \geq a_1\lambda) \tag{32}$$

$$\leq P(\frac{1}{n}\|X_j\|\|\psi'(X\beta^{oracle}) - \psi'(X\beta^{true})\| + |W^\intercal X_j| \geq a_1\lambda) \tag{33}$$

$$\leq P(\frac{1}{n}\|\psi'(X\beta^{oracle}) - \psi'(X\beta^{true})\| + |W^\intercal X_j|/\|X_j\| \geq a_1\lambda\|X_j\|^{-1}) \tag{34}$$

$$\leq P(\|\psi'(X\beta^{oracle}) - \psi'(X\beta^{true})\| + |W^\intercal X_j|/\|X_j\| \geq (ab_u n)^{1/2}a_1\lambda) \tag{35}$$

$$\leq P(b_u\|X\beta^{oracle} - X\beta^{true}\| + |W^\intercal v| \geq (ab_u n)^{1/2}a_1\lambda) \tag{36}$$

$$\leq P(b_u\|X\delta^o\| + |W^\intercal v| \geq (ab_u n)^{1/2}a_1\lambda) \tag{37}$$

where $v \in \Re^n$ is some vector with $\|v\| = 1$ as indicated in (A2) and $\delta^o = \beta^{oracle} - \beta^{true}$.

From this, using Demorgan's law and the union bound, we notice that $P(A + B \geq C) \leq P(A \geq C/2) + P(B \geq C/2)$ which can be used to further simplify

$$\phi_1 \leq P(b_u\|X\delta^o\| + |W^\intercal v| \geq a_1\lambda(ab_u n)^{1/2}) \tag{38}$$

$$\leq P(\|X\delta^o\| \geq (1/2)a_1\lambda(an/b_u)^{1/2}) + P(|W^\intercal v| \geq (1/2)a_1\lambda(ab_u n)^{1/2}) \tag{39}$$

We can then simplify both terms individually. For the first term, $P(b_u\|X\delta^o\| \geq (1/2)a_1\lambda(ab_u n)^{1/2})$, given the fact that the oracle solution and true solution have the same support, the oracle solution must be in the $\Gamma = 0$ level set of the true solution. Using similar arguments to Lemma 5, we have that $\frac{b_l}{2n}\|X\delta^o\|^2 \leq \frac{1}{n}W^\intercal X\delta^o$ From here Lemma 2 can be applied since we know $\|\beta^{oracle} - \beta^{true}\|_0 \leq s$. With some simplification this gives that $\|X\delta^o\| \leq \frac{2}{b_l}(\max_{S_{\tilde{p}}:|S_{\tilde{p}}|=s}\|\tilde{U}_{S_{\tilde{p}}}^\intercal W\|)$ Utilizing Lemma 3 with $s$ in place of $\tilde{p}$ shows that $P\left[\max_{S_{\tilde{p}}:|S_{\tilde{p}}|=s}\frac{2}{b_l}\|\tilde{U}_{S_{\tilde{p}}}^\intercal W\| \geq \frac{2}{b_l}\sigma\sqrt{s + 2\sqrt{st_1} + 2t_1}\right] \leq (\frac{pe}{s})^s \exp(-t_1)$. This is the first half of $\phi_1$ as long as $(1/2)a_1\lambda(an/b_u)^{1/2} \geq \frac{2}{b_l}\sigma\sqrt{s + 2\sqrt{st} + 2t}$ which is equivalent to the assumed condition $\lambda \geq \frac{4\sigma\sqrt{s+2\sqrt{st_1}+2t_1}}{b_l a_1(an/b_u)^{1/2}}$ Next, the second term can be easily bounded using (A2): $P(|W^\intercal v| \geq (1/2)a_1\lambda(ab_u n)^{1/2}) \leq 2\exp(-\frac{a_1^2\lambda^2 ab_u n}{8\sigma^2})$

Therefor $\phi_1 \leq (\frac{pe}{s})^s \exp(-t_1) + 2\exp(\frac{-a_1^2\lambda^2 ab_u n}{8\sigma^2})$

Next consider $\phi_2 = P(\|\beta_{\mathcal{S}}^{oracle}\|_{min} \leq a\lambda)$ First, given the assumption $\|\beta^{true]}\|_{min} > (a + 1)\lambda$ we can see that $\phi_2 = P(\|\beta_{\mathcal{S}}^{oracle}\|_{min} \leq a\lambda) \leq P(\|\beta_{\mathcal{S}}^{oracle} - \beta_{\mathcal{S}}^{true}\|_{max} > \lambda) \leq P(\|\beta^{oracle} - \beta^{true}\|_2 > \lambda) = P(\|\delta^o\|_2 > \lambda)$. Next, since we know that the support of $\beta^{oracle}$ and $\beta^{true}$ is $\mathcal{S}$, we know that $|\delta_{\mathcal{S}_c}^o| = 0 \leq 3|\delta_{\mathcal{S}}^o|$ which is the constraint for the RE condition. Therefor we know that $\frac{\|X\delta^o\|^2}{n\|\delta^o\|^2} \geq r_e$. With this and a similar line of argument as in $\phi_1$ we get that $\phi_2 \leq P(\|\delta^o\| > \lambda) \leq P(\|X\delta^o\| > \lambda\sqrt{nr_e}) \leq P(\frac{2}{b_l}(\max_{S_{\tilde{p}}:|S_{\tilde{p}}|=s}\|\tilde{U}_{S_{\tilde{p}}}^\intercal W\| > \lambda\sqrt{nr_e}) =$

$P(\max_{S_{\tilde{p}}:|S_{\tilde{p}}|=s} \left\| \tilde{U}_{S_{\tilde{p}}}^{\intercal} W \right\| > \lambda \frac{b_l \sqrt{nr_e}}{2} \geq \sigma \sqrt{s + 2\sqrt{st_2} + 2t_2}) \leq (\frac{pe}{s})^s \exp(-t_2)$ assuming that $\lambda \frac{b_l \sqrt{nr_e}}{2} \geq \sigma \sqrt{s + 2\sqrt{st} + 2t}$ which is equivalent to the condition $\lambda \geq \frac{2\sigma \sqrt{s + 2\sqrt{st_2} + 2t_2}}{b_l \sqrt{nr_e}}$

This, combined with the fact that for MCP, $a_0 = a_1 = a_2 = 1$ shows the first result.

The second result can be seen by first noting all assumptions of Theorem 1 part 2 are satisfied, where (A3) with $r_{4s}$ is implied by 7. Thus by using the same arguments as in Theorem 1 part 2 which shows that the oracle solution is unique and that the global solution is the oracle solution with some probability, since the global solution is almost surely $S^3ONC$ with $\Gamma = 0$. If we use $t = t_3$ and $t' = t_4$ we get that the probability that the global solution is not the oracle solution as $\phi_3 \leq \exp(-t_3 + \tilde{p}\ln(\frac{pe}{\tilde{p}})) + \exp(-(\tilde{p}^* + 1)(t_4 - \ln p)) \cdot \frac{1 - \exp(-(p - \tilde{p}^*)(t_4 - \ln p))}{1 - \exp(-t_4 + \ln p)}$ This combined with the first result shows that the LLA algorithm converges to the global solution in 2 iterations with probability $1 - \phi_0 - \phi_1 - \phi_2 - \phi_3$ which is the second result. $\qquad \square$

## A.2 CENTRAL LEMMAS AND THEIR PROOFS

**Lemma 1.** *Let $\beta^*$ be a $S^3ONC$ solution to 2. If assumption (A1) holds, then $P[|\beta_j^*| \notin (0, a\lambda), \forall j \in \{1, 2, ..., p\}] = 1$.*

*Proof of Lemma 1.* First, define events $\gamma_j$ and $\delta_j$ as

$$\gamma_j := \left\{ \left. \frac{\partial^2 \mathcal{Q}(\beta)}{(\partial \beta_j)^2} \right|_{\beta = \beta^*} \geq 0 \right\} \tag{40}$$

$$\delta_j := \left\{ |\beta_j^*| \in (0, a\lambda) \right\}. \tag{41}$$

First, for any given $j \in \mathcal{P}$, we solve for $P[\gamma_j \cap \delta_j]$ given our assumptions. We can start with $\left. \frac{\partial^2 \mathcal{Q}(\beta)}{(\partial \beta_j)^2} \right|_{\beta = \beta^*} \geq 0$ which gives us $1/n \sum_{i=1}^{n} \psi''(x_i^{\intercal} \beta^*) x_{i,j}^2 + P_{\lambda}''(|\beta_j^*|) \geq 0$. We can rearrange this to get $b_u \sum_{i=1}^{n} x_{i,j}^2 \geq \sum_{i=1}^{n} \psi''(x_i^{\intercal} \beta^*) x_{i,j}^2 \geq -nP_{\lambda}''(|\beta^*|) = n/a$ where we get the leftmost inequality from assumption (A1) part (i) and the rightmost equality from the definition of MCP. More concisely we have that $b_u \|X_j\|^2 \geq n/a$ which contradicts (A1) part (ii). Therefor we know $P[\gamma_j \cap \delta_j] = 0$. It should also be noted that $P[\gamma_j^c] = 0$ since $\beta^*$ satisfies $S^3ONC$ conditions. Thus, by applying Demorgan's law and then the union bound, it can be obtained that

$$0 = P[\gamma_j \cap \delta_j] = 1 - P[\gamma_j^c \cup \delta_j^c] \geq 1 - P[\gamma_j^c] - P[\delta_j^c] = 1 - P[\delta_j^c] = P[\delta_j]. \tag{42}$$

We can then apply this result to all indices to get that $P[\delta_j] = 0$ for all $j \in \{1, 2, ..., p\}$, which is the desired result. $\qquad \square$

**Lemma 2.** *Consider an arbitrary $S^3ONC$ solution $\beta^*$ to 2 with MCP. Given the event that for some integer $\tilde{p} : \|\beta^* - \beta^{true}\|_0 \leq \tilde{p}$, then $|W^{\intercal} X \delta^*| \leq \left( \max_{S_{\tilde{p}}:|S_{\tilde{p}}|=\tilde{p}} \left\| \tilde{U}_{S_{\tilde{p}}}^{\intercal} W \right\| \right) \|X \delta^*\|, \quad a.s.$*

*Where*

$$(\tilde{U}_{S_{\tilde{p}}})_{i,j} := \begin{cases} U_{S_{\tilde{p}}}, & \text{if } j \in S_{\tilde{p}} \\ 0, & \text{else} \end{cases}$$

*and $U_{S_{\tilde{p}}} \in \Re^{n \times \tilde{p}}$ is defined as in the following Thin SVD: $X_{S_{\tilde{p}}} = U_{S_{\tilde{p}}} D_{S_{\tilde{p}}} V_{S_{\tilde{p}}}$.*

*Proof.* Denote $\delta^* := (\delta_j^*) = \beta^* - \beta^{true}$, $S_{\tilde{p}} := (j : \delta_j^* \neq 0) \subseteq \mathcal{P}$, $\delta_{S_{\tilde{p}}}^* := (\delta_j^* : j \in S_{\tilde{p}})$ and $X_{S_{\tilde{p}}} := (x_{ij} : i \in \mathcal{N}, j \in S_{\tilde{p}})$. By assumption, we know that $\|\delta^*\|_0 \leq |S_{\tilde{p}}| = \tilde{p}$.

First decompose $X_{S_{\tilde{p}}}$ using Thin SVD to get $X_{S_{\tilde{p}}} = U_{S_{\tilde{p}}} D_{S_{\tilde{p}}} V_{S_{\tilde{p}}}$ where $U_{S_{\tilde{p}}} \in \Re^{n \times \tilde{p}}$. Note that since and $U_{S_{\tilde{p}}}^{\intercal} U_{S_{\tilde{p}}} = I$ we have that for any $\upsilon \in \Re^{\tilde{p}}$ we have $\left\| D_{S_{\tilde{p}}} V_{S_{\tilde{p}}} \upsilon \right\|^2 =$

$(D_{S_{\tilde{p}}}V_{S_{\tilde{p}}}\upsilon)^{\intercal}I(D_{S_{\tilde{p}}}V_{S_{\tilde{p}}}\upsilon) = \upsilon^{\intercal}V_{S_{\tilde{p}}}^{\intercal}D_{S_{\tilde{p}}}^{\intercal}U_{S_{\tilde{p}}}^{\intercal}U_{S_{\tilde{p}}}D_{S_{\tilde{p}}}V_{S_{\tilde{p}}}\upsilon = \upsilon^{\intercal}X_{S_{\tilde{p}}}^{\intercal}X_{S_{\tilde{p}}}\upsilon = \left\|X_{S_{\tilde{p}}}\upsilon\right\|^2$. therefor we can obtain that

$$
\begin{aligned}
|W^{\intercal}X\delta^*| = |W^{\intercal}X_{S_{\tilde{p}}}\delta_{S_{\tilde{p}}}^*| &\leq \left\|W^{\intercal}U_{S_{\tilde{p}}}\right\| \left\|D_{S_{\tilde{p}}}V_{S_{\tilde{p}}}\delta_{S_{\tilde{p}}}^*\right\| \\
&= \left\|U_{S_{\tilde{p}}}^{\intercal}W\right\| \left\|X_{S_{\tilde{p}}}\delta_{S_{\tilde{p}}}^*\right\| \leq \left(\max_{S_{\tilde{p}}:|S_{\tilde{p}}|=\tilde{p}} \left\|\tilde{U}_{S_{\tilde{p}}}^{\intercal}W\right\|\right) \|X\delta^*\|, \quad a.s.
\end{aligned}
\tag{43}
$$

Where

$$
(\tilde{U}_{S_{\tilde{p}}})_{i,j} := \begin{cases} U_{S_{\tilde{p}}}, & \text{if } j \in S_{\tilde{p}} \\ 0, & \text{else} \end{cases}.
$$

$\square$

**Lemma 3.** *Consider an arbitrary $S^3ONC$ solution $\beta^*$ to 2 with MCP. If (A2) holds, then for some integer $\tilde{p} \leq p$, $P\left[\max_{S_{\tilde{p}}:|S_{\tilde{p}}|=\tilde{p}} \left\|\tilde{U}_{S_{\tilde{p}}}^{\intercal}W\right\| \leq \sigma\sqrt{\tilde{p}+2\sqrt{\tilde{p}t}+2t}\right] \geq 1 - (\frac{pe}{\tilde{p}})^{\tilde{p}}\exp(-t)$. Where*

$$
(\tilde{U}_{S_{\tilde{p}}})_{i,j} := \begin{cases} U_{S_{\tilde{p}}}, & \text{if } j \in S_{\tilde{p}} \\ 0, & \text{else} \end{cases}
$$

*and $U_{S_{\tilde{p}}} \in \Re^{n\times\tilde{p}}$ is defined as in the following Thin SVD: $X_{S_{\tilde{p}}} = U_{S_{\tilde{p}}}D_{S_{\tilde{p}}}V_{S_{\tilde{p}}}$.*

*Proof.* We attempt to bound $\left(\max_{S_{\tilde{p}}:|S_{\tilde{p}}|=\tilde{p}} \left\|\tilde{U}_{S_{\tilde{p}}}^{\intercal}W\right\|\right)$. Given that we now have $W$ multiplied by a square matrix, we can apply Lemma 9. In the Lemma, let $\Sigma_u = \tilde{U}_{S_{\tilde{p}}}\tilde{U}_{S_{\tilde{p}}}^{\intercal}$. The fact that $\Sigma_u\Sigma_u = \Sigma_u$ means that $\Sigma_u$ is an idempotent matrix with $\|\Sigma_u\| \leq 1$ and $Tr(\Sigma_u) = rank(\Sigma_u) \leq rank(\tilde{U}_{S_{\tilde{p}}}) \leq rank(U_{S_{\tilde{p}}}) \leq \tilde{p}$. Lemma 9 then states that that $P\left[\left\|\tilde{U}_{S_{\tilde{p}}}^{\intercal}W\right\| \leq \sigma\sqrt{\tilde{p}+2\sqrt{\tilde{p}t}+2t}\right] \geq 1-\exp(-t)$. From this we can show that

$$
P\left[\max_{S_{\tilde{p}}:|S_{\tilde{p}}|=\tilde{p}} \left\|\tilde{U}_{S_{\tilde{p}}}^{\intercal}W\right\| \leq \sigma\sqrt{\tilde{p}+2\sqrt{\tilde{p}t}+2t}\right] \geq 1 - \binom{p}{\tilde{p}}\exp(-t) \geq 1 - (\frac{pe}{\tilde{p}})^{\tilde{p}}\exp(-t). \tag{44}
$$

Where the first inequality can seen by noting that if $\eta_k \in \Re^k$ is a sequence of i.i.d random variables and $\theta \in \Re$ is a scalar, by applying De Morgan's Law and then using the union bound, it can be obtained that $P[\max_{k\in K}\eta_k \leq \theta] = P[\bigcap_{k\in K}\eta_k \leq \theta] = 1-P[\bigcup_{k\in K}\eta_s \geq \theta] \geq 1-\sum_{k\in K}P[\eta_k \geq \theta] = 1 - |K|(1 - P[\eta_k \leq \theta])$ which yields the same inequality as in 44.

This is the desired result. $\square$

**Lemma 4.** *Consider an arbitrary $S^3ONC$ solution $\beta^*$ to 2 with MCP. Let Assumptions (A1) and (A2) hold. Given the simultaneous occurrence of (i) the event that $Q(\beta^*) \leq Q(\beta^{true}) + \Gamma$ holds for some $\Gamma \geq 0$; (ii) the event that for some integer $\tilde{p}$ : $\|\beta^* - \beta^{true}\|_0 \leq \tilde{p}$. Then for any $t > 0$, $\frac{1}{n}\|X(\beta^* - \beta^{true})\|^2 \leq \frac{4\sigma^2}{b_l^2 n}(\tilde{p}+2\sqrt{\tilde{p}t}+2t) + \frac{8}{b_l}\min\{\sum_{j\in S}P'_{\lambda}(|\beta_j^*|)|\beta_j^*|, P_{\lambda}(a\lambda)(|S| - \|\beta^*\|_0) + \Gamma\}$ holds with probability at least $1 - \exp(-t + \tilde{p}\ln(\frac{pe}{\tilde{p}}))$.*

*If in addition (A3) holds with $\tilde{p}^* \geq \tilde{p}$, then $\frac{1}{n}\|X(\beta^* - \beta^{true})\|^2 \leq \frac{8\sigma^2}{b_l^2 n}\left(\tilde{p}+2\sqrt{\tilde{p}t}+2t\right) + \frac{8}{b_l}\min\{\lambda^2(|S| - \|\beta^*\|_0)r_{\tilde{p}}^{-1}, P_{\lambda}(a\lambda)(|S| - \|\beta^*\|_0) + \Gamma\}$ holds where $r_{\tilde{p}} > 0$ for any $t > 0$ with probability at least $1 - \exp(-t + \tilde{p}\ln(\frac{pe}{\tilde{p}}))$.*

*Proof.* First, denote $\delta^* := (\delta_j^*) = \beta^* - \beta^{true}$, $S_{\tilde{p}} := (j : \delta_j^* \neq 0) \subseteq \mathcal{P}$, $\delta_{S_{\tilde{p}}}^* := (\delta_j^* : j \in S_{\tilde{p}})$ and $X_{S_{\tilde{p}}} := (x_{ij} : i \in \mathcal{N}, j \in S_{\tilde{p}})$. By assumption, we know that $\|\delta^*\|_0 \leq |S_{\tilde{p}}| = \tilde{p}$. Further, let us denote

$$\mathcal{T}_1 := \min \left\{ \sum_{j \in S} P'_\lambda(|\beta^*_j|)|\beta^{true}_j|, \ \sum_{j \in S} P'_\lambda(|\beta^*_j|)|\beta^*_j - \beta^{true}_j|, \ P_\lambda(a\lambda)(|\mathcal{S}| - \|\beta^*\|_0) + \Gamma \right\}. \quad (45)$$

We now start to define the desired bound by applying the second part of Lemma 8. The result simplified using the above definitions becomes

$$\frac{b_l}{2n} \|X\delta^*\|^2 \leq \frac{1}{n} W^\intercal X\delta^* + \mathcal{T}_1, \quad a.s. \quad (46)$$

Next, since all assumptions for Lemma 1 are satisfied, we can apply it to get

$$\frac{b_l}{2n} \|X\delta^*\|^2 \leq \frac{1}{n} \left( \max_{S_{\tilde{p}}:|S_{\tilde{p}}|=\tilde{p}} \left\| \tilde{U}^\intercal_{S_{\tilde{p}}} W \right\| \right) \|X\delta^*\| + \mathcal{T}_1. \quad (47)$$

We can then complete the square, solving for $\frac{1}{\sqrt{n}} \|X\delta^*\|$ to get

$$\frac{1}{\sqrt{n}} \|X\delta^*\| \leq \frac{1}{b_l\sqrt{n}} \max_{S_{\tilde{p}}:|S_{\tilde{p}}|=\tilde{p}} \left\| \tilde{U}^\intercal_{S_{\tilde{p}}} W \right\| + \sqrt{\left( \frac{1}{b_l\sqrt{n}} \max_{S_{\tilde{p}}:|S_{\tilde{p}}|=\tilde{p}} \left\| \tilde{U}^\intercal_{S_{\tilde{p}}} W \right\| \right)^2 + \frac{2}{b_l} \mathcal{T}_1} \quad (48)$$

$$\leq 2\sqrt{\left( \frac{1}{b_l\sqrt{n}} \max_{S_{\tilde{p}}:|S_{\tilde{p}}|=\tilde{p}} \left\| \tilde{U}^\intercal_{S_{\tilde{p}}} W \right\| \right)^2 + \frac{2}{b_l} \mathcal{T}_1}, \quad (49)$$

where the last inequality holds due to the value inside the square root being larger than the term outside. From here, squaring both sides gives us

$$\frac{1}{n} \|X\delta^*\|^2 \leq \frac{4}{b_l^2 n} \left( \max_{S_{\tilde{p}}:|S_{\tilde{p}}|=\tilde{p}} \left\| \tilde{U}^\intercal_{S_{\tilde{p}}} W \right\| \right)^2 + \frac{8}{b_l} \mathcal{T}_1. \quad (50)$$

Finally, by applying the second part of Lemma 1 we get

$$\frac{1}{n} \|X\delta^*\|^2 \leq \frac{4\sigma^2}{b_l^2 n} \left( \tilde{p} + 2\sqrt{\tilde{p}t} + 2t \right) + \frac{8}{b_l} \mathcal{T}_1. \quad (51)$$

With probability at least $1 - (\frac{pe}{\tilde{p}})^{\tilde{p}} \exp(-t)$. Thus by the definition of $\mathcal{T}_1$, the first result of the lemma has been shown.

For the second part we look to bound the central term of $\mathcal{T}_1$. We first notice (a) that since assumption (A1) holds, Corollary 4 indicates that if $\beta^*_j \neq 0 \Rightarrow |\beta^*_j| \geq a\lambda$ for all $j \in \mathcal{P}$; (b) that for this range of $\beta^*_j$, $P'_\lambda(|\beta^*_j|) = 0$; (c) that per the definition of MCP $0 \leq P'_\lambda(|\beta^*_j|) \leq \lambda$ for any $\beta^*_j \in \Re$. If we combine these observations with 45 and the definition of $\delta^*$, we can see that $\mathcal{T}_1 \leq \sum_{j \in S} P'_\lambda(|\beta^*_j|)|\delta^*|$

$\leq \lambda\sqrt{|\mathcal{S}| - \|\beta^*_{\mathcal{S}}\|_0} \cdot \|\delta^*\|$. From this, given that assumption that, for this second result (A3) holds with $\tilde{p}^* \geq \tilde{p}$, and $r_{\tilde{p}} \geq r_{\tilde{p}^*} \geq 0$ we can use (A3) part (iii) to show that $\mathcal{T}_1 \leq \lambda\sqrt{|\mathcal{S}| - \|\beta^*_{\mathcal{S}}\|_0} \cdot \frac{\|X\delta^*\|}{\sqrt{nr_{\tilde{p}}}}$.
Since this holds almost surely, it can then be combined with 47 to get

$$\frac{b_l}{2n} \|X\delta^*\|^2 \leq \frac{1}{n} \left( \max_{S_{\tilde{p}}:|S_{\tilde{p}}|=\tilde{p}} \left\| \tilde{U}^\intercal_{S_{\tilde{p}}} W \right\| \right) \|X\delta^*\| + \lambda\sqrt{|\mathcal{S}| - \|x^*_{\mathcal{S}}\|_0} \cdot \frac{\|X\delta^*\|}{\sqrt{nr_{\tilde{p}}}}. \quad (52)$$

We can then multiply by $2\sqrt{n}/b_l \|X\delta^*\|$ to get

$$\frac{1}{\sqrt{n}} \|X\delta^*\| \leq \frac{2}{b_l\sqrt{n}} \max_{S_{\tilde{p}}:|S_{\tilde{p}}|=\tilde{p}} \left\| \tilde{U}^\intercal_{S_{\tilde{p}}} W \right\| + \frac{2\lambda}{b_l\sqrt{r_{\tilde{p}}}} \sqrt{|\mathcal{S}| - \|x^*_{\mathcal{S}}\|_0}. \quad (53)$$

We then square both sides and use the rule that $(A + B)^2 \leq 2A^2 + 2B^2$ to get

$$\frac{1}{n} \|X\delta^*\|^2 \leq \left[ \frac{2}{b_l\sqrt{n}} \max_{S_{\tilde{p}}:|S_{\tilde{p}}|=\tilde{p}} \left\|\tilde{U}_{S_{\tilde{p}}}^\intercal W\right\| + \frac{2\lambda}{b_l\sqrt{r_{\tilde{p}}}} \sqrt{|\mathcal{S}| - \|x_{\mathcal{S}}^*\|_0} \right]^2 \tag{54}$$

$$\leq \frac{8}{b_l^2 n} \max_{S_{\tilde{p}}:|S_{\tilde{p}}|=\tilde{p}} \left\|\tilde{U}_{S_{\tilde{p}}}^\intercal W\right\|^2 + \frac{8\lambda^2}{b_l^2 r_{\tilde{p}}}(|\mathcal{S}| - \|x_{\mathcal{S}}^*\|_0). \tag{55}$$

Combining this with 50 yields that

$$\frac{1}{n} \|X\delta^*\|^2 \leq \frac{8}{b_l^2 n} \max_{S_{\tilde{p}}:|S_{\tilde{p}}|=\tilde{p}} \left\|\tilde{U}_{S_{\tilde{p}}}^\intercal W\right\|^2 + \frac{8}{b_l} \min\left\{ \frac{\lambda^2}{r_{\tilde{p}}}(|\mathcal{S}| - \|x_{\mathcal{S}}^*\|_0), \mathcal{T}_1 \right\}. \tag{56}$$

Finally, by applying the second part of Lemma 1 and noting from (45) that $\mathcal{T}_1 \leq P_\lambda(a\lambda)(|\mathcal{S}| - \|\beta^*\|_0) + \Gamma$ we see that

$$\frac{1}{n} \|X\delta^*\|^2 \leq \frac{8}{b_l^2 n}\sigma^2(\tilde{p} + 2\sqrt{\tilde{p}t} + 2t) + \frac{8}{b_l} \min\left\{ \frac{\lambda^2}{r_{\tilde{p}}}(|\mathcal{S}| - \|x_{\mathcal{S}}^*\|_0), P_\lambda(a\lambda)(|\mathcal{S}| - \|\beta^*\|_0) + \Gamma \right\}. \tag{57}$$

With probability at least $1 - (\frac{pe}{\tilde{p}})^{\tilde{p}} \exp(-t)$ which is the desired result. $\square$

**Lemma 5.** *Let Assumptions (A1) and (A2) hold. Consider a solution $\beta^*$ satisfying $S^3ONC$ of 2. Assume that $\mathcal{Q}(\beta^*) \leq \mathcal{Q}(\beta^{true}) + \Gamma$ holds for an arbitrary $\Gamma > 0$. For any integer $\tilde{p} : 2|\mathcal{S}| \leq \tilde{p} \leq p$ if the penalty parameters $(a, \lambda)$ satisfy that $P_\lambda(a\lambda) > \frac{\sigma^2}{2nb_l}(1 + 2\sqrt{t} + 2t) + \frac{\frac{\sigma^2}{n}|\mathcal{S}|(1+2\sqrt{t}+2t)+\Gamma b_l}{b_l(\tilde{p}+1-2|\mathcal{S}|)}$, for an arbitrary $t > 0$, then $\|\beta^* - \beta^{true}\|_0 \leq \tilde{p}$ with probability at least $1 - \exp(-(\tilde{p} + 1)(t - \ln p)) \cdot \frac{1-\exp(-(p-\tilde{p})(t-\ln p))}{1-\exp(-t+\ln p)}$.*

*Proof.* We start from the useful inequality defined in 8

$$\frac{b_l}{2n} \|X\delta^*\|^2 - \frac{1}{n}W^\intercal X\delta^* + \sum_{j\in\mathcal{S}} P_\lambda(|\beta_j^*|) \leq \sum_{j\in\mathcal{S}} P_\lambda(|\beta_j^{true}|) + \Gamma, \tag{58}$$

where $\delta^* = \beta^* - \beta^{true}$. Next, conditioning on the fact (i) that $\beta^*$ is $S^3ONC$, (ii) that all assumptions for Corollary 4 are satisfied (which implies that $P_\lambda(|\beta_j^*|) \in \{0, P_\lambda(a\lambda)\}$) and (iii) that $P_\lambda(|\beta_j^{true}|) \leq P_\lambda(a\lambda)$ we have that

$$\frac{b_l}{2n} \|X\delta^*\|^2 - \frac{1}{n}W^\intercal X\delta^* + \|\beta^*\|_0 \cdot P_\lambda(a\lambda) \leq |\mathcal{S}| \cdot P_\lambda(a\lambda) + \Gamma \tag{59}$$

Now consider an event $\mathcal{E}_1 := \{\|\beta^* - \beta^{true}\|_0 = \tilde{p} + k\}$ for an arbitrary integer $k : 1 \leq k \leq p - \tilde{p}$ Conditioning on this event, we may denote and $S_{\tilde{p}+k} \subseteq \mathcal{P}$ such that $\delta_j^* \neq 0$ for all $j \in S_{\tilde{p}+k}$. By assumption we can ensure that $|S_{\tilde{p}+k}| = \tilde{p} + k$. Also denote by $X_{S_{\tilde{p}+k}} = (x_{ij} : i \in \mathcal{N}, j \in S_{\tilde{p}+k})$ and let $\delta_{S_{\tilde{p}+k}}^* := (\delta_j^* : j \in S_{\tilde{p}+k})$. Note that conditional on $\mathcal{E}_1$, the first part of Lemma 1 (using $\tilde{p} + k$ in place of $\tilde{p}$ in Lemma 1) can be used to bound $W^\intercal X\delta^*$ in 59. Additionally, by definition $\|\beta^{true}\|_0 = |\mathcal{S}|$ and conditional on $\mathcal{E}_1$ we can apply the substitution $\|\beta^*\|_0 \geq \tilde{p} + k - |\mathcal{S}|$. This gives us

$$\frac{b_l}{2} \left\|\frac{X\delta^*}{\sqrt{n}}\right\|^2 - \frac{1}{\sqrt{n}} \left( \max_{S_{\tilde{p}+k}:|S_{\tilde{p}+k}|=\tilde{p}+k} \left\|\tilde{U}_{S_{\tilde{p}+k}}^\intercal W\right\| \right) \left\|\frac{X\delta^*}{\sqrt{n}}\right\| \leq -(\tilde{p}+k-2|\mathcal{S}|) \cdot P_\lambda(a\lambda) + \Gamma \tag{60}$$

In order for this equation to be feasible, we know that the quadratic formula must have real roots. Therefor

$$\left( \max_{S_{\tilde{p}+k}:|S_{\tilde{p}+k}|=\tilde{p}+k} \left\| \frac{\tilde{U}_{S_{\tilde{p}+k}}^{\mathsf{T}} W}{\sqrt{n}} \right\| \right)^2 - 4[b_l/2][(\tilde{p}+k-2|\mathcal{S}|)\cdot P_\lambda(a\lambda) - \Gamma] \geq 0 \qquad (61)$$

Now consier another event $\mathcal{E}_2(t) := \{\max_{|S_{\tilde{p}+k}|=\tilde{p}+k} ||U_{S_{\tilde{p}+k}}^{\mathsf{T}} W|| \leq \sigma\sqrt{\tilde{p}+k} \cdot \sqrt{1 + 2\sqrt{t} + 2t}\}$ for an arbitrary $t > 0$. Conditioning on the occurence of $\mathcal{E}_1 \cap \mathcal{E}_2(t)$ we can show, using first $\mathcal{E}_2(t)$ and then 61, that $\frac{\sigma^2(\tilde{p}+k)}{n} \cdot (1 + 2\sqrt{t} + 2t) \geq \left( \max_{|S_{\tilde{p}+k}|=\tilde{p}+k} \frac{\left\| \tilde{U}_{S_{\tilde{p}+k}}^{\mathsf{T}} W \right\|}{\sqrt{n}} \right)^2 \geq 2b_l[(\tilde{p} + k - 2|\mathcal{S}|) \cdot P_\lambda(a\lambda) - \Gamma]$ almost surely, which contradicts with the assumption on the parameters $(a,\lambda)$. This can be seen starting from our original assumption that $P_\lambda(a\lambda) > \frac{\sigma^2}{2nb_l}(1 + 2\sqrt{t} + 2t) + \frac{\frac{\sigma^2}{n}|\mathcal{S}|(1+2\sqrt{t}+2t)+\Gamma b_l}{b_l(\tilde{p}+1-2|\mathcal{S}|)} \geq \frac{\sigma^2}{2nb_l}(1+2\sqrt{t}+2t) + \frac{\frac{\sigma^2}{n}|\mathcal{S}|(1+2\sqrt{t}+2t)+\Gamma b_l}{b_l(\tilde{p}+k-2|\mathcal{S}|)}$ We can then multiply both (outer) sides by $2b_l(\tilde{p}+k-2|\mathcal{S}|)$ and rearrange to get $\frac{\sigma^2}{n}(\tilde{p}+k) \cdot (1+2\sqrt{t}+2t) < 2b_l[(\tilde{p}-2|\mathcal{S}|+k) \cdot P_\lambda(a\lambda) - \Gamma]$. Given this contradiction, we know that $P[\mathcal{E}_1 \cap \mathcal{E}_2(t)] = 0$. Therefore, again using the union bound combined with DeMorgan's law we get that $P[\mathcal{E}_1 \cap \mathcal{E}_2(t)] \geq 1 - P[\mathcal{E}_1^c] - P[\mathcal{E}_2(t)^c]$ which, can be simplified to

$$P[\mathcal{E}_2(t)^c] \geq P[\mathcal{E}_1] \qquad (62)$$

Since all assumptions of the Lemma 3 are satisfied, we can next use it to bound $P[\mathcal{E}_2(t)^c]$. By taking the compliment of the result in the second half of Lemma 1, we get, for some $t'$, that $P\left[ \max_{S_{\tilde{p}+k}:|S_{\tilde{p}+k}|=(\tilde{p}+k)} \left\| \tilde{U}_{S_{\tilde{p}+k}}^{\mathsf{T}} W \right\| \geq \sigma\sqrt{(\tilde{p}+k)+2\sqrt{(\tilde{p}+k)t'}+2t'} \right] \leq (\frac{pe}{\tilde{p}+k})^{\tilde{p}+k} \exp(-t') \leq p^{\tilde{p}+k} \exp(-t')$. If we then let $t' = (\tilde{p}+k)t$ we get $P\left[ \max_{S_{\tilde{p}+k}:|S_{\tilde{p}+k}|=(\tilde{p}+k)} \left\| \tilde{U}_{S_{\tilde{p}+k}}^{\mathsf{T}} W \right\| \geq \sigma\sqrt{\tilde{p}+k} \cdot \sqrt{1+2\sqrt{t}+2t} \right] \leq p^{\tilde{p}+k} \exp(-(\tilde{p}+k)t)$. Thus we have that $p^{\tilde{p}+k} \exp(-(\tilde{p}+k)t) \geq P[\mathcal{E}_2(t)^c]$ which can be combined with 62 to show

$$p^{\tilde{p}+k} \exp(-(\tilde{p}+k)t \geq P[\left\| \beta^* - \beta^{true} \right\|_0 = \tilde{p}+k] \quad \forall k \in \mathbb{Z}: 1 \leq k \leq p-\tilde{p}. \qquad (63)$$

With this, we can solve for our desired value

$$P\left[ \left\| \beta^* - \beta^{true} \right\|_0 \leq \tilde{p} \right] = 1 - P\left[ \left\| \beta^* - \beta^{true} \right\|_0 \geq \tilde{p}+1 \right] = 1 - \sum_{k=1}^{p-\tilde{p}} P\left[ \left\| \beta^* - \beta^{true} \right\|_0 = \tilde{p}+k \right]$$

$$\geq 1 - \sum_{k=1}^{p-\tilde{p}} \exp((\tilde{p}+k)(\ln p - t))$$

$$= 1 - \exp(-(\tilde{p}+1)(t - \ln p)) \cdot \frac{1 - \exp(-(p-\tilde{p})(t - \ln p))}{1 - \exp(-t + \ln p)}$$

$$(64)$$

Which is the desired result.

$\square$

**Lemma 6.** *Consider an arbitrary $S^3ONC$ solution $\beta^*$ to 2 with MCP. Let Assumptions (A1) and (A3) with $\tilde{p}^* \geq \tilde{p}$ hold. Assume the satisfaction of $\|\beta^* - \beta^{true}\|_0 \leq \tilde{p}$ and Event $\mathcal{E}_a(\tilde{p}) := \{\frac{1}{n}\|X(\beta^* - \beta^{true})\|^2 \leq \frac{8\sigma^2}{b_l^2 n}(\tilde{p} + 2\sqrt{\tilde{p}t} + 2t) + \frac{8}{b_l}\min\{\frac{\lambda^2}{r_{\tilde{p}}}(|\mathcal{S}| - \|\beta^*\|_0), P_\lambda(a\lambda) \cdot (|\mathcal{S}| - \|\beta^*\|_0) + \Gamma\}$. If the sub-optimality gap satisfies $\Gamma < P_\lambda(a\lambda) - \frac{\sigma^2}{b_l n}(\tilde{p} + 2\sqrt{\tilde{p}t} + 2t)$. If the minimum signal strength satisfies $\|\beta_{\mathcal{S}}^{true}\|_{\min} > \sqrt{\frac{8\sigma^2}{r_{\tilde{p}}b_l^2 n}(\tilde{p} + 2\sqrt{\tilde{p}t} + 2t) + \frac{8}{r_{\tilde{p}}b_l}\min\{\frac{\lambda^2}{r_{\tilde{p}}}|\mathcal{S}|, P_\lambda(a\lambda)|\mathcal{S}| + \Gamma\}}$ then $\beta^*$ is the oracle solution to 2.*

*If in addition we have the satisfaction of $\|\beta^{opt} - \beta^{true}\| \leq \tilde{p}$ and the event $\mathcal{E}_b(\tilde{p}) := \{\frac{1}{n}\|X(\beta^{opt} - \beta^{true})\|^2 \leq \frac{8\sigma^2}{b_l^2 n}(\tilde{p} + 2\sqrt{\tilde{p}t} + 2t) + \frac{8}{b_l}\min\{\frac{\lambda^2}{r_{\tilde{p}}}(|\mathcal{S}| - \|\beta^*\|_0), P_\lambda(a\lambda) \cdot (|\mathcal{S}| - \|\beta^*\|_0) + \Gamma\}$ then $\beta^*$ is both the oracle solution and the global solution to 2.*

*Proof.* First, let us denote $\beta^* - \beta^{true} = \delta^*$. We start by combining $\mathcal{E}_\alpha(\tilde{p})$ and (A3)iii, which is possible due to our assumption $\|\beta^* - \beta^{true}\|_0 \leq \tilde{p}$. This gives us

$$\frac{8\sigma^2}{b_l^2 n}\left(\tilde{p} + 2\sqrt{\tilde{p}t} + 2t\right) + \frac{8}{b_l}\min\{\frac{\lambda^2}{r_{\tilde{p}}}(|\mathcal{S}| - \|\beta^*\|_0), P_\lambda(a\lambda) \cdot (|\mathcal{S}| - \|\beta^*\|_0) + \Gamma\}$$
$$\geq \frac{1}{n}\|X\delta^*\|^2 \geq r_{\tilde{p}}\|\delta^*\|^2 \text{ a.s.}$$
(65)

From here if we relax $|\mathcal{S}| - \|\beta^*_{\mathcal{S}}\|_0$ to just $|\mathcal{S}|$, the definition of $\delta^*$ and note that $\|\delta^*\| \geq \|\delta^*_j\|$, we can obtain the following

$$\sqrt{\frac{8\sigma^2}{r_{\tilde{p}} b_l^2 n}\left(\tilde{p} + 2\sqrt{\tilde{p}t} + 2t\right) + \frac{8}{r_{\tilde{p}} b_l}\min\{\frac{\lambda^2}{r_{\tilde{p}}}|\mathcal{S}|, P_\lambda(a\lambda)|\mathcal{S}| + \Gamma\}}$$
$$\geq \|\beta^*_j - \beta^{true}_j\| \geq |\beta^{true}_j| - |\beta^*_j|,$$
(66)

almost surely. From this we can bound $|\beta^*_j|$ using the square root term and $|\beta^{true}_j|$, so we know that if $|\beta^{true}_j| - \sqrt{\frac{8\sigma^2}{r_{\tilde{p}} b_l^2 n}(\tilde{p} + 2\sqrt{\tilde{p}t} + 2t) + \frac{8}{r_{\tilde{p}} b_l}\min\{\frac{\lambda^2}{r_{\tilde{p}}}|\mathcal{S}|, P_\lambda(a\lambda)|\mathcal{S}| + \Gamma\}} > 0$ then $|\beta^*_j| > 0$. From this we can obtain the inequality

$$\|\beta^*_{\mathcal{S}}\|_0 \geq \sum_{j \in \mathcal{S}} \mathbb{I}\left(|\beta^{true}_j| - \sqrt{\frac{8\sigma^2}{r_{\tilde{p}} b_l^2 n}\left(\tilde{p} + 2\sqrt{\tilde{p}t} + 2t\right) + \frac{8}{r_{\tilde{p}} b_l}\min\{\frac{\lambda^2}{r_{\tilde{p}}}|\mathcal{S}|, P_\lambda(a\lambda)|\mathcal{S}| + \Gamma\}} > 0\right)$$
(67)

almost surely. We can then combine this with our minimum signal strength assumption to get that

$$\|\beta^*_{\mathcal{S}}\|_0 = |\mathcal{S}| \text{ a.s.}$$
(68)

We can combine this with equation 65, by focusing on the second part of the minimum term and noting the right side is always positive to get

$$\frac{8\sigma^2}{b_l^2 n}\left(\tilde{p} + 2\sqrt{\tilde{p}t} + 2t\right) + \frac{8}{b_l}(-P_\lambda(a\lambda)\|\beta^*_{\mathcal{S}^c}\|_0 + \Gamma) \geq 0 \text{ a.s.}$$
(69)

which can be simplified into

$$\frac{\sigma^2}{b_l n}\left(\tilde{p} + 2\sqrt{\tilde{p}t} + 2t\right) + \Gamma \geq P_\lambda(a\lambda)\|\beta^*_{\mathcal{S}^c}\|_0 \text{ a.s.}$$
(70)

thus, it can be seen that if $P_\lambda(a\lambda) > \frac{\sigma^2}{b_l n}(\tilde{p} + 2\sqrt{\tilde{p}t} + 2t) + \Gamma$ then $1 > \|\beta^*_{\mathcal{S}^c}\|_0 = 0$. This is satisfied by the assumption that $P_\lambda(a\lambda) - \frac{\sigma^2}{b_l n}(\tilde{p} + 2\sqrt{\tilde{p}t} + 2t) > \Gamma$.

Finally, because $\beta^*$ is an $S^3ONC$ solution, it has to satisfy FONC. Per 2, this means that $\beta^* \in \arg\inf\{\frac{1}{n}\sum_{i \in \mathcal{N}} \ell(\beta, x_i, y_i) + \sum_{j \in \mathcal{P}} P'_\lambda(|\beta^*_j|)|\beta_j| : \beta \in \Re^p\}$. Due to Corollary 4, we know that the penalty term goes to 0 since either $\beta^*_j = 0$ or $P'(|\beta^*_j|) = P'(|a\lambda|) = 0$. Further we know that $\beta^*_j = 0$ for all $j \in \mathcal{S}^c$. Therefore we know that

$$\beta^* \in \arg\inf\{\frac{1}{n}\sum_{i \in \mathcal{N}} \ell(\beta, x_i, y_i) : \beta \in \Re^p, \beta_j = 0, \forall j \in \mathcal{S}^c\} \text{ a.s.}$$
(71)

Given that the expression on the right is the definition of the oracle solution, we have shown the first result.

Next, Consider $\beta^{opt}$ which is the global optimal solution to 2. Given that the $S^3ONC$ conditions are necessary, $\beta^{opt}$ must be an $S^3ONC$ solution. With this fact and the assumption of $\mathcal{E}_b(\tilde{p})$, we have the same set of assumptions for $\beta^{opt}$ as we had for $\beta^*$. Thus the same sequence of arguments can be used to show that

$$\beta^{opt} \in \arg\inf\{\frac{1}{n}\sum_{i \in \mathcal{N}} \ell(\beta, x_i, y_i) : \ \beta \in \Re^p, \beta_j = 0, \forall j \in \mathcal{S}^c\} \ a.s. \tag{72}$$

Finally, per the strict convexity of our loss function as implied by (A1) we can see that the infimum of the above problem is unique. Therefor

$$\beta^* = \arg\inf\{\frac{1}{n}\sum_{i \in \mathcal{N}} \ell(\beta, x_i, y_i) : \ \beta \in \Re^p, \beta_j = 0, \forall j \in \mathcal{S}^c\} = \beta^{opt} \ a.s. \tag{73}$$

Which is the second result.

$\square$

## A.3   ADDITIONAL LEMMAS

**Lemma 7.** *The RE condition in 1 implies (A3) with $r_{4s} \geq r_e > 0$ and $\tilde{p}^* \geq 4s$.*

*Proof.* As in Lemma 1 in (Liu et al., 2017). $\square$

**Lemma 8.** *Let $\beta^*$ be a $S^3ONC$ solution to 2 Given (A1) and that $\mathcal{Q}(\beta^*) \leq \mathcal{Q}(\beta^{true}) + \Gamma$ holds for some $\Gamma \geq 0$ then*

$$\frac{b_l}{2n}\|X\delta^*\|^2 - \frac{1}{n}W^\intercal X\delta^*$$

$$\leq \min\left\{\sum_{j \in S}P'_\lambda(|\beta_j^*|)|\beta_j^{true}|, \ \sum_{j \in S}P'_\lambda(|\beta_j^*|)|\beta_j^* - \beta_j^{true}|, \ P_\lambda(a\lambda)(|\mathcal{S}| - \|\beta^*\|_0) + \Gamma\right\}, \quad a.s. \tag{74}$$

*Proof.* First, we know that $\beta^* \in \arg\min_\beta\{\sum_{i=1}^n \ell(\beta, x_i, y_i) + \sum_{j=1}^p P'_\lambda(|\beta^*|)|\beta_j|\}$ because the KKT conditions are the same as FONC which $\beta^*$ satisfies. This gives us that $\sum_{i=1}^n \ell(\beta^*, x_i, y_i) + \sum_{j=1}^p P'_\lambda(|\beta^*|)|\beta_j^*| \leq \sum_{i=1}^n \ell(\beta^{true}, x_i, y_i) + \sum_{j=1}^p P'_\lambda(|\beta^*|)|\beta_j^{true}|$. This can be used along the same lines as the level set inequality in the derivation for 8 to get $\frac{b_l}{2n}\|X\delta^*\|^2 - \frac{1}{n}W^\intercal X\delta^* \leq \sum_{j=1}^p P'_\lambda(\beta_j^*)(|\beta_j^{true}| - |\beta_j^*|)$

The first two terms of the min function are easily obtained from this. The last term can be obtained from 8 by noting that due to Corollary 4, $\beta^* \notin (0, a\lambda)$ and that $P_\lambda(a\lambda) = P_\lambda(\beta) \quad \forall \beta \geq a\lambda$. This gives us that $\frac{b_l}{2n}\|X\delta^*\|^2 - \frac{1}{n}W^\intercal X\delta^* \leq P_\lambda(a\lambda)(\mathcal{S} - \|\beta^*\|_0) + \Gamma$ Which is the final term to complete the desired result.

$\square$

**Lemma 9.** *Consider a subgaussian $\tilde{n}$-dimensional random vector $\tilde{W} \in \Re^{\tilde{n}}$ as defined in (A2). Then for any $V \in \Re^{\tilde{n} \times \tilde{n}}$ and $\Sigma_v = V^\intercal V$ then $P[\left\|V\tilde{W}\right\|^2 \leq \sigma^2 \cdot (Tr(\Sigma_v) + 2\sqrt{Tr(\Sigma_v^2)t} + 2\|\Sigma_v\|t)] \geq 1 - \exp(-t)$ for any $t > 0$ where $Tr(\cdot)$ denotes the trace of a matrix.*

*Proof.* We apply Theorem 2.1 in (Hsu et al., 2012) where our $\tilde{W}, V$ and $\Sigma_v$ are equivalent to their $x$, $A$ and $\Sigma$. Note their expectation condition is equivalent to our (A2) with $\mu = E[W] = 0$. This gives us that for all $t > 0$

$$
\begin{aligned}
P\Big[\|VW\|^2 > \sigma^2 \cdot (tr(\Sigma_v) + 2\sqrt{tr(\Sigma_v)t} + 2\|\Sigma_v\|\, t) \\
+ tr(\Sigma_v \mu \mu^\mathsf{T}) \cdot (1 + 2(\frac{\|\Sigma_v\|^2}{tr(\Sigma_v^2)}t)^{1/2})\Big] \le \exp(-t).
\end{aligned}
\tag{75}
$$

Given that $\mu = 0$, the term involving $tr(\Sigma_v \mu \mu^\mathsf{T})$ goes to zero and therefor the statement in 9 can be obtained by taking the complement of the probability bound.

$\square$

