# OpenReview forum: "Fully Polynomial-Time Randomized Approximation Schemes for Global Optimization of High-Dimensional Folded Concave Penalized Generalized Linear Models"
_ICLR.cc/2020/Conference — Reject_

### Official Review · AnonReviewer2 · 2019-10-22
**Official Blind Review #2**

**Rating:** 3

**Review:**


Summary:  The paper studies the problem of global optimization of high-dimensional sparse estimators regularized by PCP (folded concave penalty). The main result is showing that under certain conditions, with high probability, the desired global solution is an oracle stationary point satisfying the so-called S^3ONC conditions. In light of this result and an existing polynomial-time algorithm for finding an S^3ONC solution, the global solution can be recovered with polynomial computational complexity. Numerical evidence is provided to show the theoretical predictions.

Strong points:

-S1. The global optimization of non-convex (regularized) sparse learning models is a fundamental and challenging problem worth investigating.

-S2. The paper is well organized and clearly presented in general.

Weak points:

-W1. The loss function in Equation (1) is lacking in explanation. It is claimed that “traditional statistical learning schemes often resort to” such a formulation. However, it looks like only the logistic loss explicitly admits such a form while some other widely used ones such as squared/hinge/exponential loss do not. So are there any concrete statistical learning models other than logistic loss falling exactly into this framework? It could be beneficial to provide a reference, if any, to this particular problem formulation in machine learning literature.

-W2. The nominal generative model is not clearly defined. Usually, a key component of high-dimensional statistical analysis is to define a nominal statistical model for data generalization. I note the parameter vector of such a model is denoted as $\beta^{true}$ in this paper. However, the related data generation procedure seems completely missing in the statement of problem setups. Particularly, what’s the definition of the residuals W appeared in the assumption A2?

-W3. The overall novelty of theory is limited. The main result established in Theorem looks closely related to the excess risk bounds established in [Liu & Ye 2019]. Actually, provided that the risk function $L(\beta)$ has (restricted) strong convexity in $\beta$, it follows immediately based on the first excess bound in Theorem 1 of [Liu & Ye 2019] that $\|\beta^* - \beta^{true}\|$ is well bounded from above. Thus if the minimum non-zero absolute value of $\beta^{true}$ is sufficiently larger than that upper bound, then it is naturally true that $\beta^*$ shares the same supporting set as $\beta^{true}$ which in turn implies that $\beta^*$ is an oracle solution and also is globally optimal under some more stringent conditions. Unless the author(s) can justify the value-added beyond the results in [Liu & Ye 2019] as sufficient, the degree of novelty of the current theory seems fairly low given that prior work.

-W4. The proof of Theorem 1 has flaws. The proof of main result relies largely on Lemma 5 which basically bounds the cardinality $\|\beta^*-\beta^{true}\|_0$. However, when invoking Lemma 5, such a $L_0$ bound shifts to an $L_2$-norm one. This misleading point needs to be clarified.

-W5. The numerical study can be improved. The majority of the reported results is about the advantage of FCP over Lasso and ridge regression, which however is less relevant to the global recovery theory developed in the paper. I think the numerical study needs to be re-designed to put more focus on the effects of some key factors, e.g , the sample size n and the signal strength $\beta_\min$, on the global optimization performance.

-W6. Concerning the fitness to venue, although somewhat relevant,  I am not sure the main topic of this paper (i.e., statistical analysis of high-dimensional GLM) would gain significant interests in the community of ICLR. Actually in my opinion, the novelty/importance of this work is more suitable to be evaluated in a high-dimensional learning theory intensive journal or conference rather than in a DL/RL conference.

Minor issues:

- M1. Check the correctness of notation $\beta$ in Definition 1.

- M2. Equation (3): Why not directly writing out “$0 \in 1/n\sum_{i=1}^n…$”?

- M3. Statement of Theorem 1: satisfy -> satisfies. The quantities $t$ and $t’$ appeared in Theorem1 are hard to understand without further explanation.

=== update after author response ===

Thank you for the response which is helpful for clarifying some of my concerns. However, after reading the revised paper, I am still not quite convinced that the current theoretical analysis is particularly novel given the results in [Liu & Ye 2019]. Also, it seems that the noconvexity of the considered problem mainly arises from the regularization term rather than the loss term, and thus can hard to be claimed as "central to ICLR".  In view of these, I choose to maintain my assessment of this paper as borderline leaning to rejection.


**Experience Assessment:**

I have published in this field for several years.

**Review Assessment: Checking Correctness Of Derivations And Theory:**

I assessed the sensibility of the derivations and theory.

**Review Assessment: Checking Correctness Of Experiments:**

I assessed the sensibility of the experiments.

**Review Assessment: Thoroughness In Paper Reading:**

I read the paper thoroughly.

---

> ### Author Response · Authors · 2019-11-15
> **Response to Official Blind Review #2**
>
> Thank you for your comments, they have helped us to significantly improve the paper. Detailed changes below. Note ...'s were added due to char limits
>
> -W1. The loss function in Equation (1) is lacking in explanation. It is claimed that “traditional statistical learning schemes often resort to” such a formulation. However, it looks like only the logistic loss explicitly admits such a form while some other widely used ones such as squared/hinge/exponential loss do not. So are there any concrete statistical learning models other than logistic loss falling exactly into this framework? It could be beneficial to provide a reference, if any, to this particular problem formulation in machine learning literature.
>
>  Of the ones you mention, squared loss is the link function for linear regression, while exponential loss is the one for Poisson regression. The linear regression example is easiest to see by expanding .5||Y-X'B||^2, one term has no B, one is Y'X'B (as in the formulation) and the last is (X'B)^2 which is what the link function would be. There's no issue with hinge loss but I don't think it fits a canonical regression type within GLMs but certainly fits within the formulation. I also added to remark 1 on page 4 to go through some possibilities and went through 2 examples within Remark 3. I added some references for GLM use on page 1 paragraph 1 of the intro, line 6.
>
> -W2. The nominal generative model is not clearly defined. Usually, a key component of high-dimensional statistical analysis is... Particularly, what’s the definition of the residuals W appeared in the assumption A2?
>
> I give some details on this within Remark 3 on page 4 and explain the residuals within A2 on page 4.
>
> -W3. The overall novelty of theory is limited. The main result established in Theorem looks closely related to the excess risk bounds established in [Liu & Ye 2019]. Actually, provided that the risk function  has (restricted) strong convexity in beta , it follows immediately based on the first excess bound in Theorem 1 of [Liu & Ye 2019] that ||\delta*|| is well bounded from above. Thus if the minimum non-zero absolute value of beta^true  is sufficiently larger than that upper bound, then it is naturally true that beta^* shares the same supporting set as beta^true which in turn implies that beta^* is an oracle solution and also is globally optimal under some more stringent conditions. Unless the author(s) can justify the value-added beyond the results in [Liu & Ye 2019] as sufficient, the degree of novelty of the current theory seems fairly low given that prior work.
>
> It is non-trivial how the Liu and Ye 2019 paper can show oracle solution even under the RSC. Even with the sub-optimality gap bounded from above, it is unknown if the solution will fall within the restricted subset defined for the RSC. I have also added some discussion of their work in relation to our own in page 3 paragraph 3 starting from "two works".
>
> -W4. The proof of Theorem 1 has flaws. The proof of main result relies largely on Lemma 5 ... when invoking Lemma 5, such a L0 bound shifts to an L2-norm one. This misleading point needs to be clarified.
> This was a typo which has been fixed
>
> -W5. The numerical study can be improved. The majority of the reported results is about the advantage of FCP over Lasso and ridge regression, which however is less relevant to the global recovery theory developed in the paper. I think the numerical study needs to be re-designed to put more focus on the effects of some key factors, e.g , the sample size n and the signal strength , on the global optimization performance.
>
> unfortunately this is the one request i did not have time to address other than focusing more on the global optimal portion of the existing numerical results.
>
> -W6. Concerning the fitness to venue, although somewhat relevant,  I am not sure the main topic of this paper (i.e., statistical analysis of high-dimensional GLM) would gain significant interests in the community of ICLR. Actually in my opinion, the novelty/importance of this work is more suitable to be evaluated in a high-dimensional learning theory intensive journal or conference rather than in a DL/RL conference.
>
>  This work is directly related to nonconvex optimization within high-dimensional learning, something that seems central to ICLR. We have added a number of citations in line 4 of the introduction on page 1 that can help place our work more centrally within the ICLR focus given their official statement. "We consider a broad range of subject areas ... as applications in vision, speech recognition, text understanding, games, music, computational biology, and others.​"
>
> - M3. The quantities  and  appeared in Theorem1 are hard to understand without further explanation
> I have added some small intuition to remark 7 on page 5 but it should be noted that Corollary 1 is designed to explicate those relationships between important variables in a way that is impossible for the more general Theorem 1.

---

### Official Review · AnonReviewer4 · 2019-10-26
**Official Blind Review #4**

**Rating:** 3

**Review:**

=Summary=
The authors study the high-dimensional sparse estimation problems, which is one of the fundamental topics in both machine learning and optimization communities. In the literature, the folded concave penalty (FCP) methods have been shown to enjoy the strong oracle property for high-dimensional sparse estimation.  While LASSO solutions can be easily computed, such solutions do not admit unbiasedness and oracle property or require strong conditions. Therefore,  the problems with FCP have received much attentions and been studied in terms of hardness and approximability of the problems recently .
In the paper, the authors focus on the minimax concave penalty (MCP) that is a special class of FCP.
Inspire of the problem being NP-hard, they proposed pseudo-polynomial time algorithms and showed the global optimality; using the characterization of a significant subspace second-order necessary condition (S^3ONC), they showed that all local solutions within an efficiently achievable sub-level set are globally optimal. They also investigates the empirical evaluation of proposed methods.



=Significance=
In the technical view,  the part of (a) in Theorem 1 is very similar to Theorem 4 in [Hongcheng Liu, Tao Yao, Runze Li, and Yinyu Ye (Mathematical programming, 2017)]. The main result of this paper may be the part of (b) on Theorem 1.
However, the motivation why the authors wish to compute global optimization is not clear to me.
Indeed,  for FCP,  [Liu et al, 2017] showed that
1. Any local solution is a sparse estimator.
2. Any local solution satisfying a significant subspace second-order necessary condition yields a bounded error in approximating the true parameter with high probability
3. For MCP (with restricted eigenvalue condition), S^3ONC solution has oracle property.
Therefore, the global optimality is not necessarily stipulated to ensure the recovery quality.
I would like to confirm the motivation of this work.

=Missing references=
Although the topic addressed in the paper is related to the complexity of the sparse estimation problems, the following papers are not included in the references.

Huo, X. and Chen, J. Complexity of penalized likelihood estimation. Journal of Statistical Computation and Simulation, 80(7):747–759, 2010.

Bian, W. and Chen, X. Optimality conditions and complexity for non-lipschitz constrained optimization problems.

Chen, X., Ge, D., Wang, Z., and Ye, Y. Complexity of unconstrained L2 − Lp minimization. Mathematical Programming, 143(1-2):371–383, 2014.



=Questions or comments=

1. Time complexity:
Why and how the running time of the proposed algorithm can be polynomial time from pseudo-polynomial time?
Where is the time complexity of the proposed algorithm explicitly presented in the paper?
At least, the definitions of FPTAS and FPRAS and other technical terms of the complexity are missing in the paper. If the author wants to publish the paper in this conference, the definitions of them should be written in the paper, since many of readers in machine learning areas may not know the exact definitions.

2. Can the results presented in the paper extend to the case for SCAD? Since SCAD and MCP are shared with similar properties, I’d like to see more discussion on SCAD case.

3. About the problem setting:
All the inputs and parameters are assumed to be rational number?
Are there any assumption on the loss function?

4. After the definition 1,  “some independence assumptions “ or  “the same setting” are unclear. Please clarify the assumptions or setting to be self contained.



=Minor comments=

About the title: If the paper did not provide any results on SCAD, it is better to use the term "the minimax concave penalized" not "folded concave penalized"

(Page 3: “The oracle solution is a hypothetical assumes the prior knowledge on the true support set…” is grammatically wrong?)

Remark 1: remove  one “gradient"

Remark 4: S^3ONC

References: Cun-Hui Zhang, Tong Zhang, et al.
-> add "and", remove "et al."

References: Cun-Hui Zhang et al. => remove et al.

Page 18:  delete the memo “ move to intro somehwere?”



**Experience Assessment:**

I have read many papers in this area.

**Review Assessment: Checking Correctness Of Derivations And Theory:**

I assessed the sensibility of the derivations and theory.

**Review Assessment: Checking Correctness Of Experiments:**

I did not assess the experiments.

**Review Assessment: Thoroughness In Paper Reading:**

I read the paper at least twice and used my best judgement in assessing the paper.

---

> ### Author Response · Authors · 2019-11-15
> **Response to Official Blind Review #4**
>
> Thank you for your comments, I have tried to address each of them and they have significantly improved the paper.
>
> >In the technical view,  the part of (a) in Theorem 1 is very similar to Theorem 4 in [Hongcheng Liu, Tao Yao, Runze Li, and Yinyu Ye (Mathematical programming, 2017)]. The main result of this paper may be the part of (b) on Theorem 1.
> >However, the motivation why the authors wish to compute global optimization is not clear to me.
> Indeed,  for FCP,  [Liu et al, 2017] showed that
> >1. Any local solution is a sparse estimator.
> >2. Any local solution satisfying a significant subspace second-order necessary condition yields a bounded error in approximating the true parameter with high probability
> >3. For MCP (with restricted eigenvalue condition), S^3ONC solution has oracle property.
> >Therefore, the global optimality is not necessarily stipulated to ensure the recovery quality.
> >I would like to confirm the motivation of this work.
>
> In a broader sense: Nonconvex optimization in learning problems are becoming more and more common. Many of those problems are NP-hard and thus may seem computationally challenging in large-scale applications. This result at least identifies perhaps one of the first examples that the worst-case complexity can be overly pessimistic; FPRAS exists for globally solving some of those strongly NP-hard problems.
>
> In a narrower sense: the definition of the estimators defined with MCP-based regularization, technically speaking, stipulates global optimality. It is because of the NP-hardness, that our theories focus on local solutions to gain computational plausibility. Now that we know that global solutions are FPRAS exists, future analysis on MCP-based GLMs can start directly from analyzing global solutions. This will impact problems such as statistical testing, e.g., when designing log-likehood ratio tests, usually the global optimality is stipulated.
>
> > =Missing references=
> >Although the topic addressed in the paper is related to the complexity of the sparse estimation problems, the following papers are not included in the references.
>
> >Huo, X. and Chen, J. Complexity of penalized likelihood estimation. Journal of Statistical Computation and Simulation, 80(7):747–759, 2010.
>
> >Bian, W. and Chen, X. Optimality conditions and complexity for non-lipschitz constrained optimization problems.
>
> >Chen, X., Ge, D., Wang, Z., and Ye, Y. Complexity of unconstrained L2 − Lp minimization. Mathematical Programming, 143(1-2):371–383, 2014.
>
> These have been added, with both compexity papers cited in the intro on page 1 around line 11 while the optimality conditions paper is cited as another example of nonconvex regularized minimization on page 2 at the very bottom
>
> >1. Time complexity:
> >Why and how the running time of the proposed algorithm can be polynomial time from pseudo-polynomial time?
> >Where is the time complexity of the proposed algorithm explicitly presented in the paper?
> >At least, the definitions of FPTAS and FPRAS and other technical terms of the complexity are missing in the paper. If the author wants to publish the paper in this conference, the definitions of them should be written in the paper, since many of readers in machine learning areas may not know the exact definitions.
>
> This was a mistake in our language, we have clarified our time complexities in a variety of locations and include the definition of FPRAS within Defintion 4 on page 6. The time complexity of the proposed algorithm was also added within Remark 11 on page 6.
>
> >2. Can the results presented in the paper extend to the case for SCAD? Since SCAD and MCP are shared with similar properties, I’d like to see more discussion on SCAD case.
>
> No, they cannot, the title has been changed as a result.
>
> >3. About the problem setting:
> >All the inputs and parameters are assumed to be rational number?
> >Are there any assumption on the loss function?
>
> no assumptions are made about rational or irrational numbers within the analysis. This work applies to both. The assumption on the loss function is in A1 on page 4.
>
> >4. After the definition 1,  “some independence assumptions “ or  “the same setting” are unclear. Please clarify the assumptions or setting to be self contained.
>
> This has been clarified and fleshed out in the first paragraph above Preliminaries on S3ONC on page 4. Suffice it to say it does apply to our setting.
>
> >About the title: If the paper did not provide any results on SCAD, it is better to use the term "the minimax concave penalized" not "folded concave penalized"
>
> This has been changed as suggested
>
> >(Page 3: “The oracle solution is a hypothetical assumes the prior knowledge on the true support set…” is grammatically wrong?)
>
> this has been corrected
>
> >Remark 1: remove  one “gradient"
>
> fixed
>
> >Remark 4: S^3ONC
>
> corrected
>
> >References: ... (running out of space)
>
> these have been fixed
>
> >Page 18:  delete the memo “ move to intro somehwere?”
>
> this has been fixed

---

### Official Review · AnonReviewer1 · 2019-10-31
**Official Blind Review #1**

**Rating:** 8

**Review:**

This is a theory heavy paper. My only concern is relevance for this conference, but other than that the results are interesting and useful.

It seems the authors have focused on a particular case of folded convex penalty (Minimax Concave Penalty or MCP --- the authors should provide the full form of the abbreviation before using it). The major contribution of this work is the analysis, while the algorithmic setup can be borrowed from previous works (e.g. Liu/Ye 2019).

Remark 10 is not clear to me. Why is the assumption that \beta^\star has a lower value than \beta^{Lasso} reasonable ? Also, the assumption is to hold almost surely, which I am assuming is over possible instantiation of the randomized scheme ? In that case, this assumption seems very strong in general, unless I am missing something.

I would suggest authors expand the paper to be self-sufficient, instead of referring to other works (e.g. Algorithm 1 in Liu/Ye 2019, and proof of Lemma 9 ). Please proof read e.g. above equation 62.

It would be nice if the authors also provide discussions on how just the sample complexity being large enough is sufficient for assumptions of Theorem 1. I had to spend a lot of time to understand the relationship and intuition wrt \Gamma implying from sample complexity. That should add to the readability of the paper.





**Experience Assessment:**

I have published one or two papers in this area.

**Review Assessment: Checking Correctness Of Derivations And Theory:**

I assessed the sensibility of the derivations and theory.

**Review Assessment: Checking Correctness Of Experiments:**

I carefully checked the experiments.

**Review Assessment: Thoroughness In Paper Reading:**

I read the paper at least twice and used my best judgement in assessing the paper.

---

> ### Author Response · Authors · 2019-11-15
> **Response to Official Blind Review #1**
>
> Thank you for your comments, I the paper is significantly improved as a result. Details on specific changes below.
>
> >It seems the authors have focused on a particular case of folded convex penalty (Minimax Concave Penalty or MCP --- the authors should provide the full form of the abbreviation before using it)
>
> We have changed the title and fixed our abbreviation usage
>
> >The major contribution of this work is the analysis, while the algorithmic setup can be borrowed from previous works (e.g. Liu/Ye 2019).
>
> We added Page 3 paragraph 3 (not including cut off one) starting from "two works" describing the comparison between notable works and our own including the Liu/Ye 2019 in order to address this.
>
> >Remark 10 is not clear to me. Why is the assumption that \beta^\star has a lower value than \beta^{Lasso} reasonable ? Also, the assumption is to hold almost surely, which I am assuming is over possible instantiation of the randomized scheme ? In that case, this assumption seems very strong in general, unless I am missing something.
>
> We have updated Remark 11 (was 10) on page 6 to clarify this. The idea is that both FPTAS’s are described as always being initialized at \beta^{Lasso}. So unless our S3ONC algorithm ascends the gradient of the MCP problem, it is not possible to get a worse FCP objective than that of the solution to the LASSO problem. Given that our algorithm is a FPTAS, there are no issues using Lasso as a subroutine within it. It should also be noted that the comparison is for the MCP regularized objective, not the non regularized one.
>
> >I would suggest authors expand the paper to be self-sufficient, instead of referring to other works (e.g. Algorithm 1 in Liu/Ye 2019, and proof of Lemma 9 ).
>
> This change has been made, algorithm 1 on page 6 has been fleshed out while the proof of Lemma 9 is more descriptive though still requires their result within the proof.
>
> >Please proof read e.g. above equation 62.
>
> This has been fixed
>
> >It would be nice if the authors also provide discussions on how just the sample complexity being large enough is sufficient for assumptions of Theorem 1. I had to spend a lot of time to understand the relationship and intuition wrt \Gamma implying from sample complexity. That should add to the readability of the paper.
>
> This has been added to remark 8 (was 7) on page 5

---

### Decision · Program_Chairs · 2019-12-19

**Decision:**

Reject

**Comment:**

Thanks for your detailed feedback to the reviewers, which clarified us a lot in many respects.
However, the novelty of this paper is rather marginal and given the  high competition at ICLR2020, this paper is unfortunately below the bar.
We hope that the reviewers' comments are useful for improving the paper for potential future publication.